# Targeting an allosteric site in dynamin-related protein 1 to inhibit Fis1-mediated mitochondrial dysfunction

Luis Rios [1,4], Suman Pokhrel [1,4], Sin-Jin Li [2,3], Gwangbeom Heo[1], Bereketeab Haileselassie[2] & Daria Mochly-Rosen [1] ✉

The large cytosolic GTPase, dynamin-related protein 1 (Drp1), mediates both physiological and pathological mitochondrial fission. Cell stress triggers Drp1 binding to mitochondrial Fis1 and subsequently, mitochondrial fragmentation, ROS production, metabolic collapse, and cell death. Because Drp1 also mediates physiological fission by binding to mitochondrial Mff, therapeutics that inhibit pathological fission should spare physiological mitochondrial fission. **P110**, a peptide inhibitor of Drp1-Fis1 interaction, reduces pathology in numerous models of neurodegeneration, ischemia, and sepsis without blocking the physiological functions of Drp1. Since peptides have pharmacokinetic limitations, we set out to identify small molecules that mimic **P110**'s benefit. We map the **P110**-binding site to a switch I-adjacent grove (SWAG) on Drp1. Screening for SWAG-binding small molecules identifies **SC9**, which mimics **P110**'s benefits in cells and a mouse model of endotoxemia. We suggest that the SWAG-binding small molecules discovered in this study may reduce the burden of Drp1-mediated pathologies and potentially pathologies associated with other members of the GTPase family.

Mitochondrial dysfunction during inflammatory cell stress has been identified as an important driver of diseases with unmet clinical needs, such as sepsis[1–3], neurodegeneration[4,5], and ischemic injury[6–10]; rampant cell stress and inflammation can lead to widespread metabolic collapse and organ failure. Damaged mitochondrial components are directly involved in pro-inflammatory signaling and healthy mitochondria are in turn damaged by inflammatory signals, creating a robust pro-inflammatory vicious cycle[11]. When cell stress is high or prolonged, mitochondria undergo excessive fission leading to mitochondrial fragmentation, reduced mitochondrial membrane potential, increased ROS, and reduced ATP production, which triggers apoptosis[12–14]. Therefore, pharmacologic inhibition of pathological mitochondrial fission is a promising strategy to quell overactive inflammation and prevent mitochondrial dysfunction in a broad range of cell types and diseases[15–17].

The primary driver of mitochondrial fragmentation is Drp1, a large cytosolic GTPase that translocates from the cytosol to the outer mitochondrial membrane, where it oligomerizes and mediates mitochondrial fission through a constricting loop[18]. Drp1 is recruited to the mitochondria by partner proteins Fis1, Mff, MiD49, and MiD51[19]. Although Drp1 is an attractive drug target, its activity may not be dispensable. Drp1-Mff-mediated fission is important for maintaining mitochondrial quality and inhibition of this interaction is cytotoxic[20]. In vivo, Drp1-Mff inhibition pharmacologically or genetically accelerates the pathology associated with neurodegenerative disease[20,21]. In contrast, Drp1-Fis1 interaction mediates pathological conditions; a rationally designed heptapeptide inhibitor of Drp1-Fis1 interaction, **P110**, selectively inhibits pathological and not physiological mitochondrial fission[22]. **P110** treatment reduces the pathologies associated with a variety of diseases, including

[1]Department of Chemical and Systems Biology, Stanford University School of Medicine, Stanford, CA, USA. [2]Department of Pediatrics, Stanford University School of Medicine, Stanford, CA, USA. [3]Bachelor Program of Biotechnology and Food Nutrition, National Taiwan University, Taipei City, Taiwan. [4]These authors contributed equally: Luis Rios, Suman Pokhrel. ✉e-mail: Mochly@stanford.edu

Huntington's disease (HD)[23-27], amyotrophic lateral sclerosis (ALS)[26,28], Alzheimer's disease (AD)[26,29], Parkinson's disease (PD)[22,30,31], scrapie[32], cardiac ischemia[33], stroke[34], inflammatory bowel disease[35], and sepsis[36-38], in mouse and patient-derived cell models. Importantly, **P110** has no apparent toxicity in mice even after five months of sustained delivery at 5 mg/kg/day[28]. Fis1 may also have physiological roles through its interaction with other proteins; however, Drp1-Fis1 interaction is specifically induced during inflammatory cell death[22,26,39]. In addition to the lack of phenotype with pharmacologic inhibitors under basal conditions, Fis1 knock-down with siRNA does not have an impact on basal mitochondrial fission[40], and Fis1-KO has only a minor phenotype[41]. In contrast, Drp1-Mff interaction is found in both basal and stress conditions and Mff-KO has a profound effect on basal mitochondrial fission resulting in severely elongated mitochondria[41]. Mff may also recruit Drp1 during cell death and some genetic models indicate that Mff knock-down can reduce cell death in culture and infarct size during ischemia-reperfusion injury[41,42]. However, inhibition of Mff is not a viable strategy to reduce inflammatory cell death since pharmacologic and genetic perturbation of Mff is toxic in several cell-types and in vivo[20,21]. This is because Mff-Drp1 interaction governs essential functions under basal conditions whereas Fis1-Drp1 interaction does not.

We have previously shown that different proteins that interact in an inducible manner often share short sequences of homology that represent competitive sites of inter/intra-molecular interactions[43]. The **P110** heptapeptide (DLLPRGS) was rationally designed from such a short homology sequence between hDrp1 (49-55; DLLPRGT) and hFis1 (60-66; ELLPKGS)[22]. Cross-linking of the **P110** heptapeptide to TAT$_{47-57}$ enabled peptide delivery into cells in culture and in vivo. The term **P110** hereafter refers to the TAT conjugated **P110** heptapeptide. Although **P110** has shown great benefit in animal models of numerous human diseases, its translation to a clinical intervention has been limited because **P110** is susceptible to serum and cellular proteases and is not orally bioavailable.

We seek to identify small molecules that could mimic **P110**'s therapeutic effect. However, the feasibility of a small molecule mimetic is uncertain, since **P110**'s binding site is unknown and protein-protein interactions are often large, shallow surfaces that are difficult to inhibit with small molecules[44]. Here, we elucidate **P110**'s mechanism of action and map **P110**'s binding site. We then identify a small molecule that mimics **P110**. Our study demonstrates that small molecule inhibitors of protein-protein interactions can be generated for Drp1 and that the **P110**-binding site is an unrecognized allosteric backdoor for Drp1 inhibition that is selective for pathological fission. Our work also suggests that this allosteric strategy may be generalized to other members of the GTPase protein family.

## Results

### P110, a guanosine nucleotide-dependent uncompetitive inhibitor of Drp1
In addition to inhibiting Drp1-Fis1 interaction, we found that **P110** causes partial inhibition of Drp1 GTPase activity in vitro[22], indicating a direct binding of **P110** to Drp1. However, **P110**'s inhibitory kinetics were unknown. A TAT-free cyclic DLLPRGS (**cP110**; a form with increased peptide stability) decreased both Vmax and Km of Drp1 GTPase activity (Fig. 1a–c), indicating that **P110** is an uncompetitive inhibitor that stabilizes the GTP-bound form of Drp1. We corroborated this finding using a BODIPY-TMR labeled **P110** (BDY-**P110**); BDY-**P110** had an increased affinity for Drp1 (Kd ± SEM) in the presence of the non-hydrolysable GTP analog, GMPPCP (apparent Kd = 18 nM ± 23 nM) and GTP (apparent Kd = 58 nM ± 46 nM), as well as with GDP (apparent Kd = 219 nM ± 219) and it did not bind apo-Drp1 at the concentrations tested (Fig. 1d). Thus, **P110** engages an allosteric site, perhaps adjacent to the GTP-binding site that is induced by nucleotide binding to Drp1.

### Drp1 switch I-adjacent groove (SWAG), a prominent allosteric site near the P110-homology region
The **P110** homology sequence in Drp1 ($_{49}$DLLPRGT$_{55}$) is derived from the first half of switch I ($_{49}$DLLPRGTGIVT$_{59}$), an unstructured loop that is involved in GTP hydrolysis in a variety of GTPases[45] (Fig. 1e,f; GDP.AlF4 bound Drp1 structure, PDB: 3W6P)[46]. Overlaying this nucleotide-bound structure with the apo structure (PDB: 4BEJ)[47] suggested that the endogenous **P110** region swings by ~9 Å, making backbone contacts with amino acids R53 and G54 to the GTP α-phosphate (Supplementary Fig. 1a). The structural shift was analogous in the GMPPCP-bound Drp1 structure, 3W6O (Supplementary Fig. 1b)[46]. This swing also forms a trans-loop contact between the G54 backbone and S35 sidechain of the P-loop, an opposing mobile loop that envelops GTP when bound (Supplementary Fig. 1c). D49 and R53 in the **P110** region make loop-stabilizing contacts at the N-terminal side of switch I (Supplementary Fig. 1d,e). When switch I engages GTP, it opens a prominent groove flanking switch I, which we termed the SWAG (cyan in Fig. 1f,g). The Site Finder algorithm in Molecular Operating Environment (MOE) suggested SWAG as the third largest pocket in the GTPase domain with a positive score for the propensity of ligand binding[48] (PLB: 1.08), and, as expected, the nucleotide-binding pocket was ranked first with the largest volume and most favorable propensity of the ligand-binding score (PLB: 3.97). We next performed Essential Site Scanning Analysis[49] on PDB: 3W6P structure to identify the regions on the protein that can function as druggable sites that would modulate Drp1 GTPase activity. As expected, some residues lining the nucleotide-binding site had high druggable z-scores (magenta, Fig. 1h). However, the residues that encompass SWAG had the highest z-scores and this site appeared as the most druggable region in the GTPase domain. This analysis suggests that SWAG-binding small molecules could allosterically modulate the GTPase activity and function of Drp1.

The SWAG site is defined by a subset of unstructured residues from G$_{47}$-V$_{58}$ (including switch I/**P110**-derived sequence), the alpha helical residues from F$_{104}$-E$_{111}$ (G-domain α3), and the G5 element, including residues R$_{247}$-V$_{258}$ (Supplementary Fig. 1f). The SWAG-related residues are evolutionarily identical from humans to zebrafish (Supplementary Fig. 2a) but are much less conserved in other members of the human GTPase superfamily, with the most similarity to the dynamin-2 SWAG (82% identical; Supplementary Fig. 2b).

### P110 binding depends on SWAG residues D49 and R53 for regulating Drp1 GTPase activity
To assess the role of SWAG as a potential **P110**-binding site, we mutated the two charged residues of the **P110**-derived sequence in Drp1; D49 and R53 were mutated to an alanine. This double mutant Drp1 had increased Vmax and Km (Supplementary Fig. 3a–d), suggesting decreased stability of the GTP-bound form and increased turnover, an observation that is supported by the loop-stabilizing interactions found in the crystal structure of Drp1 (Supplementary Fig. 1d,e; Supplementary Fig. 3a–d). Importantly, this Drp1 double mutant was insensitive to **cP110**; at 1 μM **cP110**, Drp1's Km and Vmax were not significantly affected (Fig. 1i,j relative to Fig. 1b,c).

### In silico screening of the CoCoCo library identifies drug-like SWAG ligands
We proceeded with a structure-based virtual screen of the Commercial Compound Collection (CoCoCo)[50], a 6.9 million compound library in multi-conformer format, for small molecules that may bind SWAG. Using an eight-feature pharmacophore and excluded volume criteria to avoid clashes with the protein, we identified 604 compounds between 260 Da and 675 Da that satisfied at least 7 of the 8 pharmacophore features and scored below S = −5.9 (Fig. 2a). We named these potential hits SWAG compounds or **SC1-SC604** in order of their predicted affinity. After removing potentially problematic compounds

with PAINS and Brenk filters using the SwissADME tool[51], we selected the top 3 compounds, **SC1**, **SC3**, and **SC9** for biochemical evaluation (Fig. 2b; Supplementary Tables 1,2; Quality control in Supplementary Fig. 4–6).

### SWAG ligands mimic P110's biochemical activity in vitro

Like **P110, SC1, SC3**, and **SC9** act as uncompetitive inhibitors of WT Drp1; at 1 μM they reduced both Km and Vmax of Drp1 (Fig. 3a–c). The

D49A Drp1 mutant was inhibited by cP110, **SC1**, and **SC3**, but not by **SC9** (Fig. 3d–f). Like **cP110, SC1** and **SC9** inhibitory activities were significantly reduced in the D49A/R53A double Drp1 mutant (Fig. 3g–i). **SC3** inhibitory activity became non-significant with the double mutation although the decrease in Vmax was close to significance ($p = 0.071$), suggesting that **SC3** may have some activity on the double mutant Drp1.

To examine if the compounds bind to the **P110**-binding site, we conducted a competitive Drp1-binding assay, by assessing the amount

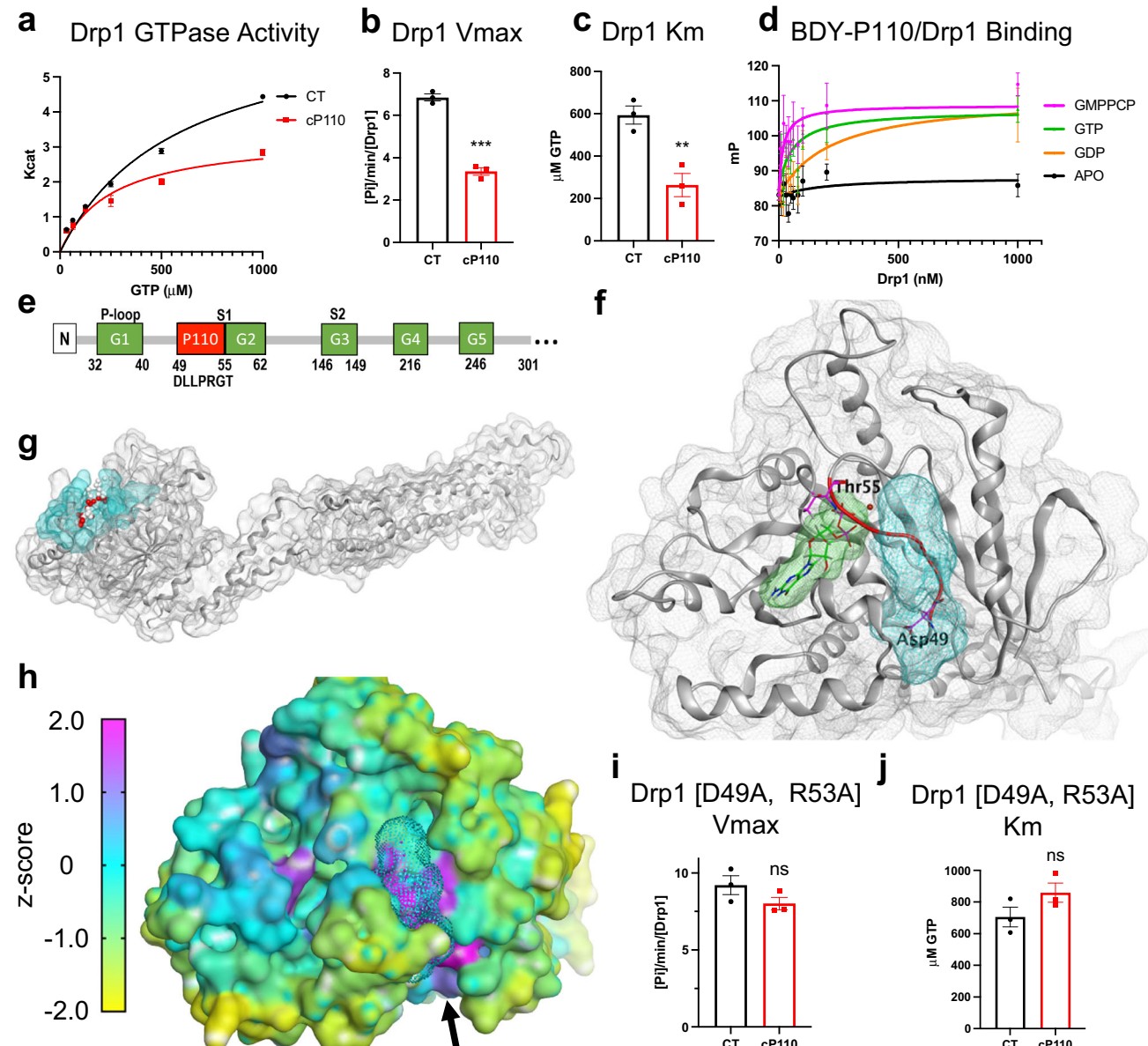

**Fig. 1 | Drp1 switch I-adjacent groove (SWAG) and P110 binding. a–c** GTPase kinetics using 100 nM recombinant Drp1 in the presence of 1 μM cP110 or vehicle (CT). $n = 3$ independent experimental averages with 3 technical replicates each. Kcat is expressed as the production of inorganic phosphate [Pi] μM/min/[Drp1 μM]. **d** Fluorescence polarization affinity assay with 10 nM BODIPY-TMR-P110 in the presence of 200 μM nucleotides ($n = 6$ replicates over 2 independent experiments). The following are the estimated Kd (±SEM) for P110-Drp1 interaction in the presence of GMPPCP (18 nM ± 23 nM), GTP (58 nM ± 46 nM), GDP (219 nM ± 219 nM), APO (n/a). **e** Linear map of Drp1 indicating the relative locations (not to scale) of the key catalytic residues (G1-5, green), switch I (S1), switch II (S2), and the endogenous P110 site (D49-T55, red). **f** Relative position of SWAG (cyan) in the overall structure of Drp1. **g** Relative positions of the nucleotide binding site (green), SWAG (cyan), and the endogenous P110 sequence DLLPRGT (red) in the GTPase domain of Drp1.

**h** Drp1 GTPase head domain with residues color coded by the z-score result from the Essential Site Scanning Analysis (ESSA). A high positive z-score (magenta) indicates the presence of a druggable functional site while an intermediate (cyan) or negative (yellow) score indicates the absence of a druggable functional site. Two sites were identified; the left purple site is the GTP binding site, and the right purple site (indicated by the black arrow) is the SWAG (cyan dots space fill). **i, j** Km and Vmax of Drp1 D49A/R53A double mutant treated with 1 μM cP110 or vehicle (CT). $n = 3$ independent experimental averages with 3 technical replicates each. All error bars show the mean with SEM. A two-tailed $t$-test was used to compare groups and the $p$-value is indicated by stars (ns $p > 0.05$, **$p < 0.01$, and ***$p < 0.001$). Source data including all statistics (degrees of freedom, $p$ values, effect sizes, and confidence intervals) are provided in the Source Data file.

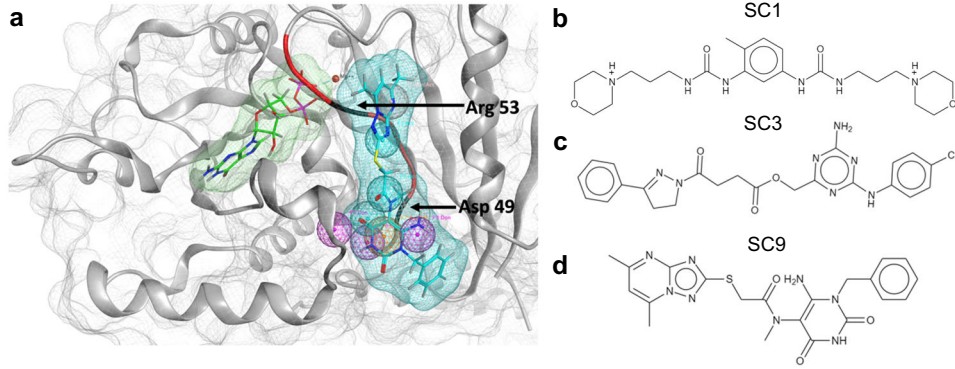

**Fig. 2 | Structure based virtual screen of the Drp1 SWAG. a** Relative positions of the nucleotide binding site (green), SWAG (cyan), and the endogenous P110 sequence DLLPRGT (red) in the GTPase domain of Drp1. Pharmacophore features of SWAG are shown as colored spheres. The Magenta sphere indicates the donor, the cyan sphere indicates the acceptor, the orange sphere indicates the aromatic or pi ring and the light orange sphere indicates the donor or acceptor feature. One of the conformations of SC9 satisfying 7 out of 8 pharmacophore features of SWAG is shown. **b–d** Top 3 hit compounds from the virtual screen [SC1 (**b**), SC3 (**c**), and SC9 (**d**)] which were selected for further investigation based on predicted binding affinity and predicted properties using the SwissADME tool. Red denotes oxygen atoms and blue denotes nitrogen atoms.

of labeled **cP110** that co-sediments with oligomerized Drp1. In the vehicle condition (CT), 1 nM BDY-**cP110** binds Drp1-GMPPCP oligomers and is pulled from the supernatant during centrifugation, resulting in decreased fluorescence in the supernatant. Pre-treating with a large excess of unlabeled **cP110** (10 μM) blocks all **P110**-specific BDY-**cP110** binding to Drp1, resulting in an increased BDY signal in the supernatant (Fig. 3j, **cP110**) and acts as a positive control in our assay. All three compounds also reduced the co-sedimentation of BDY-**cP110** with recombinant Drp1, whereas Mdivi-1[52], a weak Drp1 inhibitor, did not (Fig. 3j) and thus served as a negative control in this assay. This result demonstrates that **P110** binds to Drp1 oligomers and that **SC1**, **SC3**, and **SC9** compete with this binding.

Since Dnm2 has the most similar SWAG to Drp1 (82% homology; Supplementary Fig. 2b), we tested the ability of these compounds to inhibit the GTPase activity of dynamin 2, Dnm2. **SC1** inhibited the GTPase activity of both Drp1 and Dnm2, while **cP110**, **SC3**, and **SC9** inhibited the GTPase activity of Drp1 only (Fig. 3k vs. 3l).

## Like P110, SC9 reduces LPS-induced mitochondrial dysfunction in H9c2 cells

To determine if the compounds mimic the bioactivity of **P110**, we treated cultured H9c2 cardiomyocytes with the bacterial endotoxin lipopolysaccharide (LPS) to cause pathological mitochondrial fission and dysfunction[36]. As before, we found mitochondrial fragmentation after 16 h of LPS treatment (Fig. 4a, Control vs. LPS + Vehicle). **P110** and **SC9** both blocked the LPS-induced decrease in mitochondrial size and form factor, **SC1** caused more modest effects, and **SC3** exacerbated the LPS-induced decrease in mitochondrial size and form factor relative to LPS alone (Fig. 4a,b; Supplementary Fig. 7a). We binned the cells into three classes according to mitochondrial morphology phenotype: fragmented, intermediate, and fused (Fig. 4b,c). After LPS treatment, the percent of cells in the fused state decreased by ~50% (from 38% to 20% for CT vs. LPS+Veh) and the percent of cells in the fragmented state increased by over five-fold (from 4% to 22% for CT vs. LPS + Veh). **P110** and **SC9** completely prevented the LPS-induced decline in cells with fused mitochondria; the percent of cells with fused mitochondria was 37% (LPS + **P110**), 36% (LPS + **SC9**), and 38% (no LPS). **SC1** caused only a marginal recovery in fused state cells (26%; LPS + **SC1**). **SC9** was more effective in decreasing the percentage of cells with fragmented mitochondria. **SC9** reduced the percent of fragmented cells from 22% (LPS + Veh) to 9% (LPS + **SC9**), relative to 4% in the absence of LPS; **P110** and **SC1** caused only a slight decrease in the fragmented phenotype from 22% to 18%. In contrast to the other compounds, **SC3** greatly

exacerbated the LPS effect, decreasing the proportion of cells with fused mitochondria from 20% to 3% and more than doubling the percent of cells with fragmented mitochondria from 22% to 55% (Fig. 4c), suggesting that **SC3** is cytotoxic.

As reported previously[26,36], LPS decreased mitochondrial membrane potential as measured by TMRM (Fig. 4d; Supplementary Fig. 7b, Control vs. Veh). Both **P110** and **SC9** prevented the drop in mitochondrial membrane potential, whereas **SC1** and **SC3** did not (Fig. 4d). After LPS treatment, the number of cells with high TMRM signal decreased by more than two-fold [from 34% (CT) to 15% (LPS+Veh)] and both **P110** and **SC9** increased this number to 30% (LPS + **P110**) and 41% (LPS + **SC9**). In contrast, **SC1** and **SC3** had no beneficial effect on the LPS-induced drop in mitochondrial membrane potential (Fig. 4e). Correlating TMRM signal and mitochondrial size in single cells demonstrated that LPS induced a decrease in mitochondrial membrane potential in cells with both fused and fragmented mitochondria (Supplementary Fig. 7c–g). This two-dimensional analysis demonstrates a similar correlation in LPS + **P110** and LPS + **SC9**-treated cells to control-treated cells, lack of benefit by **SC1** treatment, and toxicity mediated by **SC3**. Therefore, in the next experiments, we focused our efforts on **SC9**.

LPS-treated cells with dysfunctional mitochondria produce more ROS[26,36] as measured by MitoSOX fluorescence (Fig. 4f; Supplementary Fig. 7h, Control vs. Veh) and both **P110** and **SC9** inhibited this rise in mitochondrial ROS levels (Fig. 4f, g; Supplementary Fig. 7h). After LPS treatment, the number of cells with high ROS was more than doubled [from 18% (CT) to 42% (LPS + Veh)] and the number of cells in the low-ROS state decreased from 18% (CT) to 2% (LPS + Veh). **P110** and **SC9** decreased the number of high ROS cells to levels close to baseline (no LPS) levels (Fig. 4g).

## Like P110, SC9 inhibits Drp1-Fis1 interaction in cells

To confirm that **SC9** works by the same mechanism as **P110**[22], we next determined whether **SC9** inhibits Drp1 association with the mitochondria and Drp1-Fis1 interaction following LPS treatment. Using immunocytochemistry, we first determined the number of large oligomeric Drp1 species on the mitochondria in single cells. Large Drp1 oligomers, which are the active species of Drp1[18], appear as foci in immunocytochemistry, and their number and size can be quantified by the fluorescence signal (Supplementary Fig. 8a). This method does not measure total Drp1 levels, since lower order monomers, dimer, and tetramers are not detected. Furthermore, we restricted our analysis to detect only polymerized Drp1 (the Drp1 state that mediates

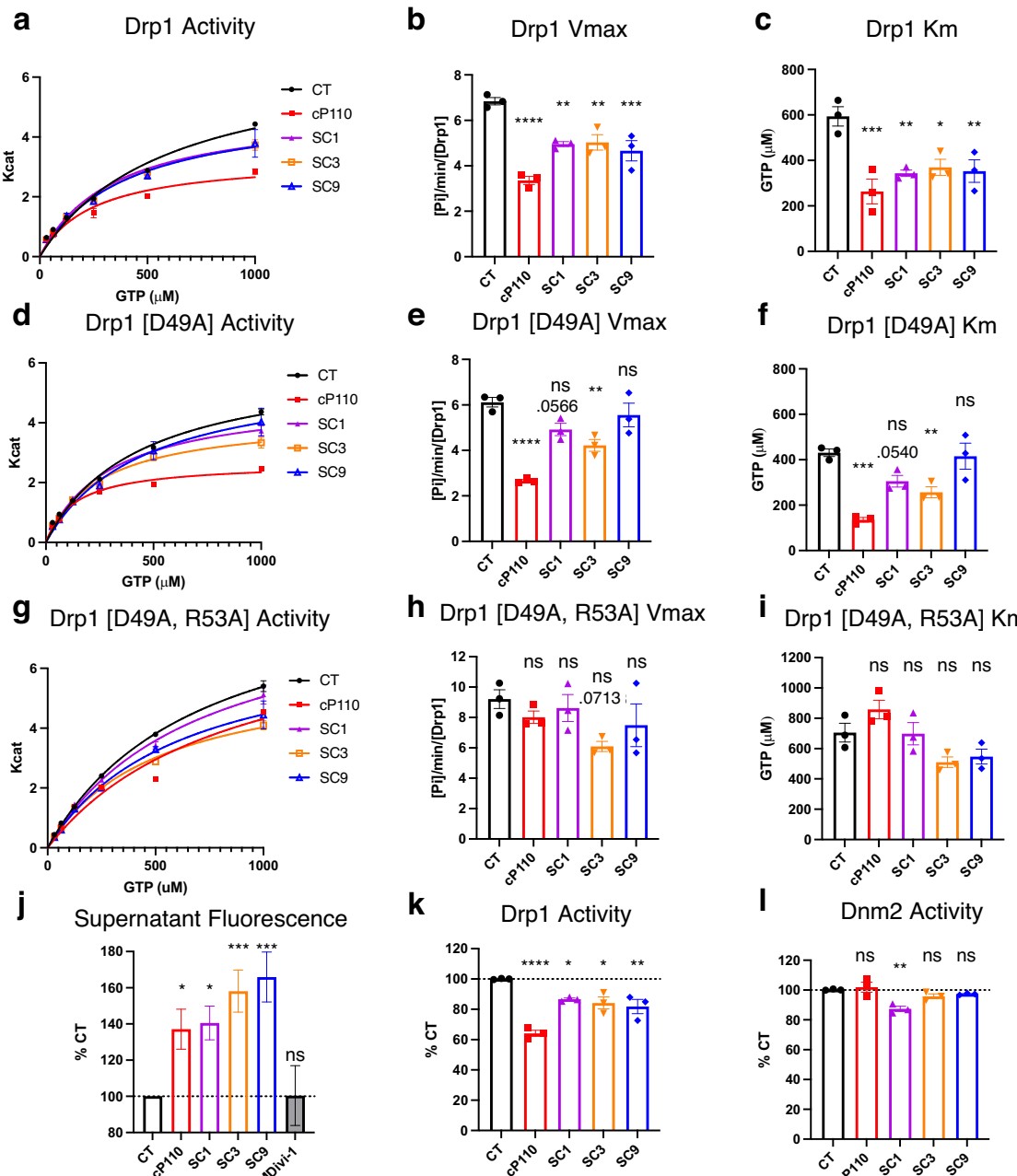

**Fig. 3 | SC1, SC3, and SC9 mimic P110 kinetic properties and Drp1 binding in vitro. a–i**, GTPase activity of 100 nM recombinant Drp1 WT (**a–c**), D49A (**d–f**), and D49A/R53A (**g–i**). Drp1 mutants treated with 1 μM compounds or vehicle control (CT). Michaelis-Menten curves generated from 3 independent experimental averages with 3 technical replicates each to estimate Km and Vmax. **j** Inhibition of 1 nM BODIPY-TMR-cP110 co-sedimentation with recombinant Drp1 (oligomerized with 200 μM GMPPCP) in the presence of vehicle (CT), unlabeled cP110, SC1, SC3, SC9, or mdivi-1 at 10 μM; data are shown as the percent of unbound fluorescent peptide signal that remains in the supernatant due to competition by 10 μM unlabeled compound ($n = 14$ independent reactions with averages of 3 technical

replicates per reaction; $n = 3$ reaction averages of 3 technical replicates per reaction for mdivi-1). **k, l** GTPase activity (normalized to control) of Drp1 (**k**) and Dnm2 (**l**) ($n = 3$ independent experimental averages with 3 technical replicates each). All error bars show mean with SEM and one-way ANOVA with Dunnett's multiple comparisons test (each group against CT, two-tailed test). *P*-value results indicated by stars (ns $p > 0.05$, *$p < 0.05$, **$p < 0.01$, ***$p < 0.001$, and ****$p < 0.0001$). Non-significant *p*-values < 0.1 are written above the bar. Source data including all statistics (degrees of freedom, *p* values, effect sizes, and confidence intervals) are provided in the Source Data file.

mitochondrial fission) on the surface of the mitochondria. LPS increased the amount of mitochondria-associated Drp1 foci by about two-fold and both **P110** and **SC9** reduced the signal to below the level of control (Fig. 5a,b; Supplementary Fig. 8a,b). To confirm that the increase in Drp1 oligomers was not due to a change in the total Drp1 protein levels, we conducted Western blot analyses on whole cell lysates and observed no difference in total Drp1 protein levels after treatment (Fig. 5c; Supplementary Fig. 8c). This indicates that **P110** and

**SC9** block stress-induced Drp1 oligomerization, a process that occurs on the mitochondrial surface.

Using a proximity ligation assay, we next measured Drp1-Fis1 and Drp1-Mff association in single cells. Consistent with a pathological stimulus that causes increased mitochondrial fragmentation, LPS increased the interaction between Drp1 and Fis1 and both **P110** and **SC9** blocked this increased interaction, with **SC9** reducing the signal to below control levels (Fig. 5d–f). **SC9** decreased the Drp1-Fis1

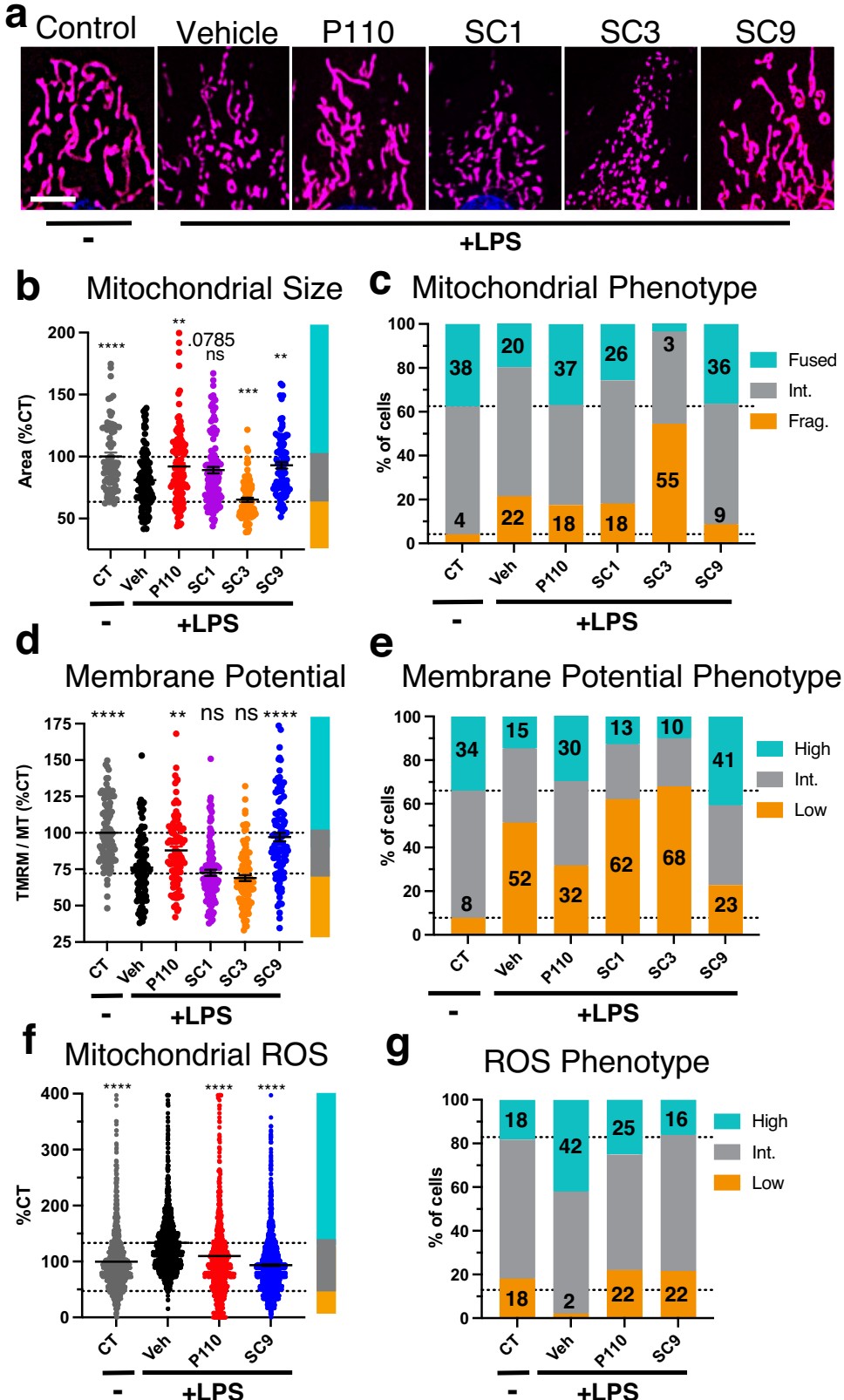

interaction to a greater extent than **P110** (Fig. 5d–f). Using a similar analysis to Fig. 4, median PLA signal cutoffs were used to bin the cells into three phenotypes: high, intermediate, and low PLA signal (Fig. 5f). After LPS treatment, the number of cells with high Drp1-Fis1 interaction more than doubled [from 26% (CT) to 67% (LPS + Veh)] and the number of cells with low levels of Drp1-Fis1 interaction decreased [from 21%

(CT) to 2% (LPS+Veh)]. **P110** and **SC9** completely blocked the LPS effect; the number of high Drp1-Fis1 cells decreased from 67% (LPS + Veh) to 28% (LPS + **P110**) and to 14% (LPS + **SC9**; see also Supplementary Fig. 8d). In contrast, Drp1 interaction with Mff, which mediates physiological fission[20], was not significantly affected by LPS, nor by **P110** and **SC9** (Fig. 5g–i and Supplementary Fig. 8e). Overall, these data

**Fig. 4 | Compound treatment effect on LPS-induced mitochondrial fragmentation and dysfunction in H9c2. a** Representative images of mitochondrial staining with MitoTracker Deep Red FM and nuclei staining with blue Hoechst stain after LPS (2 μg/ml) and 2 μM compound co-treatment for 16hrs (white scale bar = 10 μm). Cells were imaged for 20 min. For the dot plots (**b, d, f**) data normalized to the control mean and dotted lines and colored bars showing size cut-off for the phenotype classification in plotted in (**c, e, g**). For the stacked bar graphs in (**c, e, g**) dotted lines show control proportions, the lower set of numbers indicating the percent of cells in the lower bin and the numbers on top indicating the percent of cell in the upper bin. **b, c** Manually segmented single cell average mitochondrial size normalized to control (CT). CT: $n = 72$ cells; Veh: $n = 107$ cells; P110: $n = 114$ cells; SC1: $n = 121$ cells; SC3: $n = 88$ cells; SC9: $n = 91$ cells; cells examined over 2 independent experiments. **d, e** Manually segmented single cell average TMRM signal

(TMRM integrated density/MitoTracker 2D area). CT: $n = 103$ cells; Veh: $n = 103$ cells; P110: $n = 88$ cells; SC1: $n = 103$ cells; SC3: $n = 100$ cells; SC9: $n = 101$ cells; cells examined over 2 independent experiments. **f, g** Auto-segmented single cell Mito-SOX signal (MitoSOX median intensity/MitoTracker 2D area). CT: $n = 1558$ cells; Veh $n = 1593$ cells: P110: $n = 1027$ cells; SC9: $n = 1928$ cells; cells examined over 3 independent experiments. Error bars show mean with SEM and one-way ANOVA with Dunnett's multiple comparisons test (each group against Veh., two-tailed test). $P$-value results indicated by stars (ns $p > 0.05$, *$p < 0.05$, **$p < 0.01$, ***$p < 0.001$, and ****$p < 0.0001$). Non-significant $p$-values $< 0.1$ are written above bar. Single cell manual segmentation was conducted by an observer blinded to the experimental conditions. Source data including all statistics (degrees of freedom, $p$ values, effect sizes, and confidence intervals) are provided in the Source Data file.

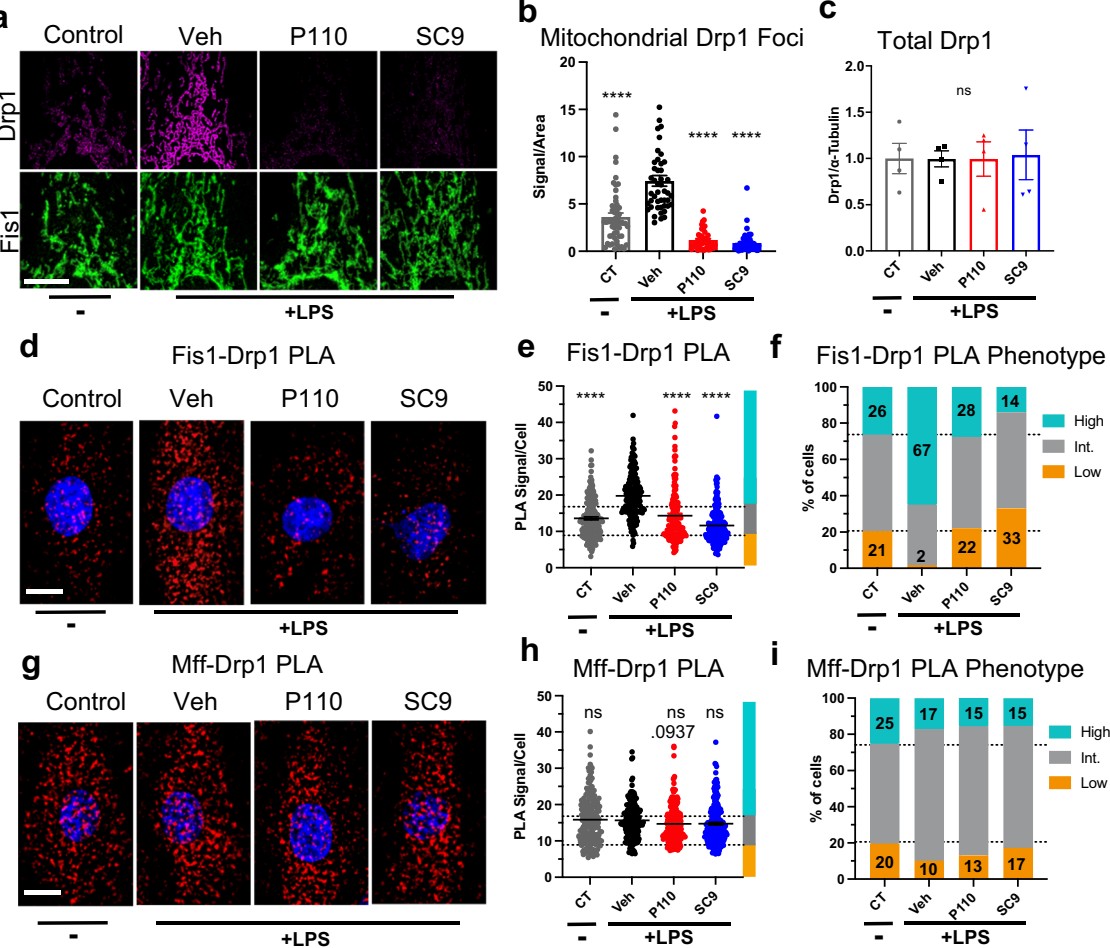

**Fig. 5 | LPS-induced mitochondrial Drp1 translocation and protein-protein interaction with Fis1 and Mff. a** Representative maximum intensity projection of Drp1 immuno-staining (magenta) collected from 20 z-stack images taken with a 100x objective. Fis1 staining (green, bottom) used as mask to quantify total mitochondrial Drp1 (magenta, top). **b** Quantification of total mitochondrial Drp1 foci (integrated density) per mitochondrial area with each data point representing a single cell (CT: $n = 55$ cells; Veh: $n = 46$ cells; P110: $n = 44$ cells; SC9: $n = 60$ cells; cells examined over 2 independent experiments). **c** Western blot quantification results showing the total levels of Drp1 after LPS and compound treatment ($n = 4$ lysates from 4 independent experiments). Drp1-Fis1 (**d–f**), and Drp1-Mff (**g–i**) PLA signal per cell after LPS treatment. **d, g** Representative images of PLA stain. **e, f, h, i** Single cells manually segmented by blinded observer and total PLA signal (integrated

density) per cell area is plotted (cells examined over 2 independent experiments). Fis1-Drp1 PLA (**e, f**) CT: n = 175 cells; Veh: $n = 179$ cells; P110: $n = 167$ cells; SC9; $n = 200$ cells. Drp1-Mff PLA (**h, i**) has: CT: $n = 188$ cells; Veh: $n = 203$ cells; P110: $n = 213$ cells; SC9: $n = 209$ cells. Error bars show mean with SEM and one-way ANOVA with Dunnett's multiple comparisons test (each group against Veh., two-tailed test). $P$-value results indicated by stars (ns $p > 0.05$ and ****$p < 0.0001$). Non-significant $p$-values $< 0.1$ are written above bar. All white scale bars are 10 μm. Source data including all statistics (degrees of freedom, $p$ values, effect sizes, and confidence intervals) are provided in the Source Data file.

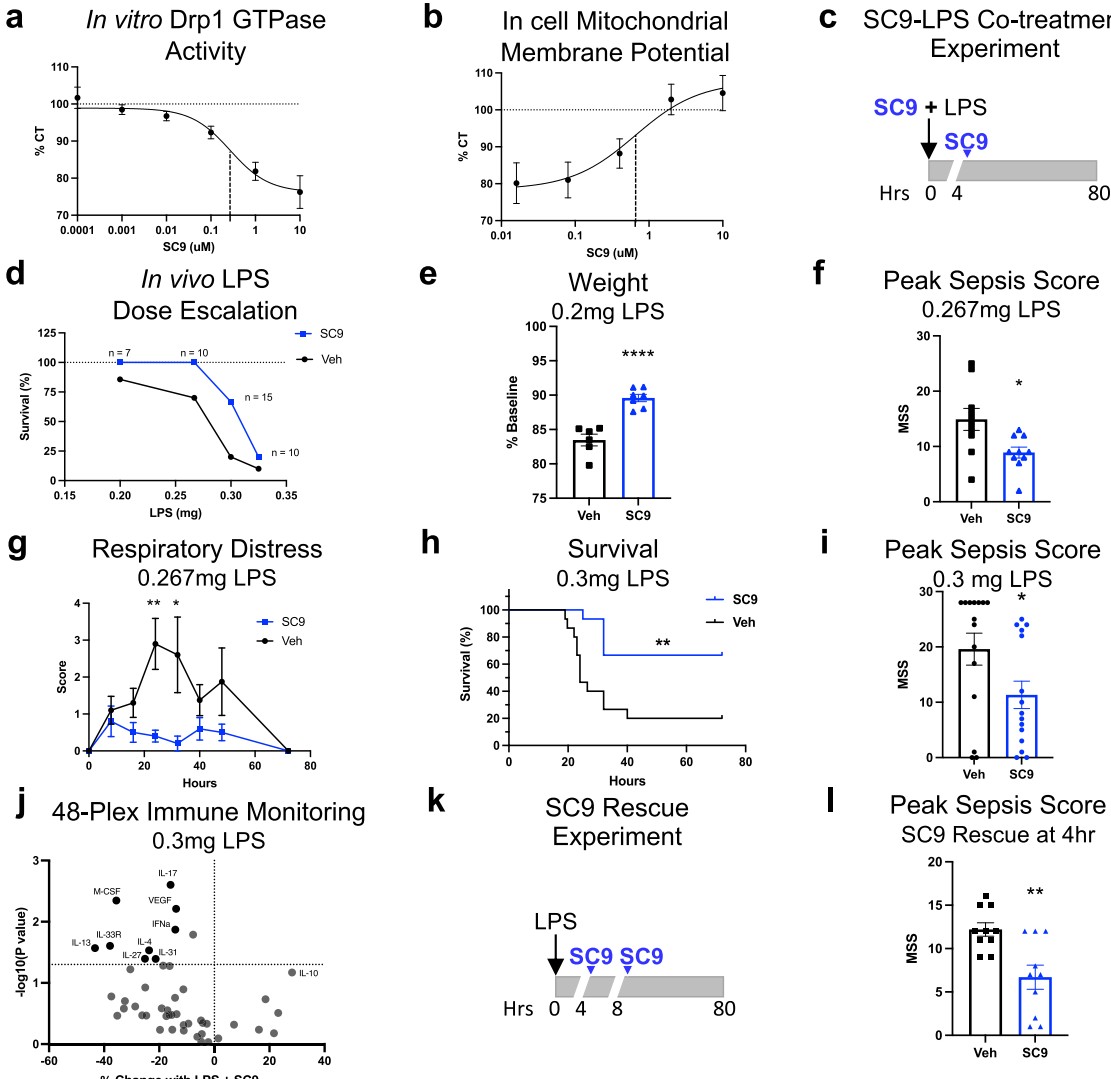

**Fig. 6 | In vitro and in cell SC9 dose-response and in vivo efficacy of SC9 in LPS-induced endotoxemia mouse model. a** Inhibition dose-response of SC9 on recombinant Drp1 GTPase activity (μM phosphate/hr) in the presence of 1000 μM GTP normalized to Veh control (dotted line); 3 independent experiments with 3 replicates each; IC50 ± SEM = 269 nM ± 59 nM. **b** SC9 dose-response in inhibiting LPS-induced mitochondrial membrane depolarization in H9c2 cells. TMRM signal quantified with a 96-well microplate reader. Data are normalized to the no-LPS control which is shown as dotted line 4 independent experiments with 5 replicates per experiment; EC50 ± SEM = 751 nM ± 290 nM. **c** Experimental protocol of LPS-induced endotoxemia mouse model with SC9 co-treatment with SC9 re-dosed at the 4-h time point. **d** Summary of survival and the number of animals per group in four LPS-induced endotoxemia mouse experiments with increasing doses of LPS co-administered with DMSO (Veh, black) or SC9 (10 mg/kg I.P x 2; blue). 7-week old female Balb/c mice were used for all the experiments. **e** Mouse body mass (% baseline) 120 h after injection of 0.2 mg LPS (Veh: $n$ = 6 mice; SC9 $n$ = 7 mice). **f** Cumulative murine sepsis score (MSS) at peak sepsis symptoms (24 h) after 0.267 mg LPS treatment ($n$ = 10 mice per group). **g** Mean respiratory distress score

assessed every 8 h after 0.267 mg LPS treatment ($n$ = 10 mice per group at 24 h). **h** Survival curve after 0.3 mg LPS with Log-rank (Mantel-Cox) test to determine significance ($n$ = 15 mice per group at start of experiment). **i** Cumulative murine sepsis score (MSS) at peak sepsis symptoms (24 h) after 0.3 mg LPS treatment ($n$ = 15 mice per group). **j** Volcano plot showing average % change in SC9-treated animals and $p$-value of each analyte in the ProcartaPlex panel (Veh: $n$ = 3 mice; SC9 $n$ = 4 mice). **k** Experimental protocol of LPS-induced endotoxemia mouse model with SC9 rescue treatment delivered 4-h and 8-h after 0.3 mg LPS injection ($n$ = 10 mice per group). **l** Cumulative murine sepsis score (MSS) at peak sepsis symptoms (18 h) after 0.3 mg LPS treatment and SC9 delivered as a rescue treatment 4-h after LPS injection ($n$ = 10 mice per group). For mouse data, a two-tailed $t$-test was used to determine $p$-value. Error bars show mean with SEM. $P$-value results indicated by stars (ns $p$ > 0.05, *$p$ < 0.05, **$p$ < 0.01, and ****$p$ < 0.0001). MSS was assessed while blinded to the experimental conditions. Source data including all statistics (degrees of freedom, $p$ values, effect sizes, and confidence intervals) are provided in the Source Data file.

indicate that both **P110** and **SC9** block mitochondrial translocation of Drp1 to the mitochondria by selectively inhibiting Drp1-Fis1 interaction.

## SC9 has nM potency
Although identified by an in silico screen, **SC9** is quite potent; the IC50 of the partial GTPase inhibition in vitro was 270 nM ± 60 nM (IC50 ± SEM; Fig. 6a and dot plot in Supplementary Fig. 9a). In cells, the dose-response curve shows that **SC9** has a similar EC50 of 750 nM ± 290 nM (IC50 ± SEM) when measuring inhibition of LPS-induced mitochondrial depolarization after 24-h (Fig. 6b and Supplementary Fig. 9b;).

## SC9 rescues mice from LPS-induced endotoxemia
We previously reported that **P110** reduces mortality in mice given a bolus intraperitoneal injection of LPS[36]. We first determined the suitability of **SC9** for in vivo studies. Using >90% pure **SC9**, as confirmed by

high resolution LC-ESI/MS and NMR (Supplementary Fig. 6c), we first intraperitoneally injected mice with increasing quantities of **SC9** to rule out overt toxicity; injected mice showed no decrease in body weight (Supplementary Fig. 9c), or clinical symptoms as determined by the murine sepsis score[53] (MSS) at doses up to 50 mg/kg. We next assessed **SC9** levels following a single intraperitoneal injection (20 mg/kg). At 15-min post-injection, **SC9** plasma levels reached 5.8 μg/ml (or 12.4 μM) which is ~17x the EC50 in culture, indicating that we achieved a pharmacologically relevant dose. **SC9** did not significantly penetrate the brain when measured after 15 min; the brain to plasma ratio of **SC9** was about 2% (Supplementary Fig. 9d). **SC9** was also eliminated quickly; after 4 h plasma levels of **SC9** were 0.2% of peak levels (Supplementary Fig. 9e) indicated a plasma half-life of <0.5 h. In LPS-treated mice injected twice with **SC9** (10 mg/kg) at 0 h and 4 h post-injection, we found 1.6% of peak **SC9** levels at 16 h after the first injection, suggesting a longer half-life in LPS-treated mice (Supplementary Fig. 9f).

Next, to determine if **SC9** resulted in on-target or off-target toxicity in vivo, we conducted standard safety toxicology screening. Female and male 7-weeks old Balb/c mice were injected with 50 mg/kg **SC9** and observed for 3 days. Mice were then sacrificed and serum samples were screened for 11 biomarkers for hematologic, hepatic, renal, and musculoskeletal damage. No significant difference in any of the clinical observations or serum measures were noted by an observer blinded to the experimental conditions (Supplementary Fig. 9g–q).

We then conducted a blinded dose escalation experiment with LPS (0.2 mg to 0.325 mg per mouse or 10 to 16.6 mg/kg) with co-administration of **SC9** at 10 mg/kg and a re-dose of **SC9** 4 h after LPS injection in 7-week-old female Balb/c mice (Fig. 6c). Although the different LPS doses were optimal for different measurements, **SC9** improved mouse survival at all LPS doses tested (Fig. 6d, Supplementary Fig. 10). With the lowest dose of LPS (0.2 mg), **SC9** reduced the body weight recovery time at 120–140 h; **SC9** also delayed the increase in murine sepsis score (MSS) at 4 h after this dose of LPS, but not at later timepoints (Fig. 6e, and Supplementary Fig. 10a,b). The effect of **SC9** treatment *vs.* vehicle treatment on mouse body mass could not be determined at time points later than 24 h, because the most severely affected mice die and fall out of the analysis, resulting in survivor bias at LPS doses of 0.3 mg and higher (Supplementary Fig. 10g,j). **SC9** reduced the total MSS at the 24-h time point by ~40% with 0.267 mg LPS (Fig. 6f, Supplementary Fig. 10e). Most importantly, **SC9** dramatically reduced the occurrence of critical symptoms such as respiratory distress, reducing peak scores from 2.9 to 0.4 (−86%) in mice challenged with 0.267 mg LPS (Fig. 6g). 0.3 mg LPS was the optimal dose for measuring survival differences; **SC9** increased mice viability after LPS treatment by more than three-fold (from 20% to 67%, Fig. 6h) and decreased MSS by ~40% (Fig. 6i, Supplementary Fig. 10h). Lower LPS doses also showed reduced death, but since the LPS-induced death was lower, these effects were not significant (Supplementary Fig. 10c, f vs. i). Challenging mice with 0.325 mg LPS appears to overwhelm the therapeutic effect of **SC9**, showing only a small increase in viability and a non-significant difference in MSS score (*p* = 0.0843, Supplementary Fig. 10j–l).

To explore the inflammatory factors that may drive the difference between **SC9** and vehicle-treated mice, we collected plasma 4 h after 0.3 mg LPS injection and carried out the mouse immune monitoring 48-plex ProcartaPlex panel. Compared to vehicle-treated mice, levels of inflammatory factors were down ~19% in **SC9**-treated mice relative to vehicle-treated LPS (Fig. 6j, Supplementary Fig. 10m, Supplementary Fig. 11a). Using a volcano plot, we displayed the factors that were most significantly down: IL-13 (−43%), IL-33R (−38%), M-CSF (−36%), IL-27 (−25%), IL-4 (−24%), IL-31 (−21%), IL-17 (−16%), IFN-α (−14%) and VEGF (−14%). The anti-inflammatory IL-10 increased (+28%; *p* = .067) although this effect was non-significant (Fig. 6j).

The disease pathophysiology of LPS-induced endotoxemia in vivo involves many cell types. To determine if **SC9** inhibits LPS-induced

mitochondrial dysfunction in other cell-types, we repeated the assays in three other cell types. Like cardiac H9c2 cells, HEK293 cells (human embryonic kidney cells) and RAW 264.7 cells (mouse macrophage-like cells) treated with **SC9** or **P110** produced less mitochondrial ROS in the presence of LPS (Supplementary Fig. 11b,c). ROS in **SC9** treated cells was reduced to the level of control. To determine if the effect is specific to LPS or if it is generalizable, we tested the ability of **SC9** to inhibit mitochondrial ROS induced by 3 h of peroxide treatment in mouse embryonic fibroblasts (MEFs). Peroxide induced a 2X increase in mitochondrial ROS and **SC9** decreased ROS production to the level of control (Supplementary Fig. 11d, left). We tested Drp1 knock-out (KO) MEFs to determine if this ROS production is Drp1-dependent and if **SC9** has an effect on Drp1 KO cells. Drp1 KO cells displayed no increase in ROS production from peroxide and there was no effect observed with **SC9** treatment indicating that the ROS response is Drp1-dependent (Supplementary Fig. 11d, right). This indicates that **SC9** blocks mitochondrial dysfunction induced by multiple pathological agents and in multiple cell types, including a human cell line. Future studies will determine the mechanisms of disease pathophysiology and its inhibition in vivo.

To better model clinical endotoxemia, we assessed whether **SC9** can rescue mice from endotoxemia after symptom onset. We conducted an LPS-induced endotoxemia mouse experiment with **SC9** delivered 4-h after LPS administration (Fig. 6k, protocol scheme). We found that at 4-hours post LPS treatment, inflammatory markers were universally elevated with 6/48 inflammatory markers up over 100x, 20/48 markers up over 10x, 7/48 markers up over 5x, and 14/48 markers up over 2x (see Supplementary Fig. 11e). MIP-2, IL-6, CCL2, CCL4, CXCL1, and IL-22 were up over 200x from baseline indicating a strong induction of inflammation before treatment. IL-2 and IL-7 were not significantly elevated. **SC9** (20 mg/kg) rescue treatment 4-hours post-LPS treatment (0.3 mg) caused an increase in survival (although not significant due to lower overall death) and did not affect body weight (Supplementary Fig. 11f, g). In addition, the mice exhibited a significant reduction in disease symptoms as determined by the murine sepsis score and this was comparable to the benefit of **SC9** pre-treatment (MSS; Fig. 6l vs. 6i;); symptom severity plateaued after **SC9** administration (dashed line and blue arrow in Supplementary Fig. 11h), resulting in a 45% decrease in the peak sepsis score (from 12.2 to 6.7; *p* = 0.0029; Fig. 6l). These results indicate that **SC9** can still be efficacious when administered after the induction of the inflammatory response.

## Discussion

In this study, we identified an allosteric site in large GTPases, which we have termed the switch I-adjacent groove (SWAG). We identified a compound, **SC9**, which binds to SWAG and inhibits the GTPase activity of Drp1 and not of dynamin-2, the GTPase with the most similar SWAG. It is unlikely that **SC9** will inhibit other large GTPases such as MX1, OPA1, or MFN1/2 since the SWAG site in these proteins is divergent from Drp1; the SWAG region of these proteins is only 47%, 29%, and 29% identical to Drp1, respectively. However, the SWAGs present in these other large GTPases could also potentially bind small molecules or regulate proteins. Studies investigating the SWAGs in other large GTPases could lead to the discovery of novel allosteric functional sites in these proteins and to small molecules that regulate them.

Much effort has focused on the GTP-binding site near switch I[45], but the large SWAG site, exposed by GTP binding, has not yet been explored in any GTPase. The SWAG could represent an inducible protein-protein interaction site for protein partners to directly modulate GTP affinity and GTPase activity through switch I/GTP stabilization. For Drp1, this SWAG site may represent the Fis1-selective binding site or may allosterically expose the Fis1-binding site elsewhere on Drp1; these two possibilities are under investigation. Regardless, we find molecules that bind to SWAG inhibit Drp1-Fis1 interaction and are

effective and selective inhibitors of pathological and not physiological mitochondrial fission.

The **P110** sequence in hDrp1 (amino acids 49-55) is within switch I, and it forms a mobile loop alternating between occupying SWAG and stabilizing GTP. We provide several lines of evidence demonstrating that **P110**, which we identified as a selective inhibitor of Drp1-Fis1 interaction and mitochondrial dysfunction[22], shares a common mechanism with **SC9**: (1) both **P110** and **SC9** exert similar uncompetitive inhibitory kinetics on Drp1 GTPase activity; (2) their effect is lost when two amino acids associated with the SWAG, D49, and R53, are mutated to alanine; (3) **SC9** competitively inhibits **P110** binding to Drp1; (4) both **SC9** and **P110** inhibit LPS-induced Drp1-Fis1 association in cells, without affecting Drp1 association with its physiological mitochondrial adaptor, Mff; (5) they both inhibit mitochondrial fragmentation and dysfunction in cells treated with LPS; (6) they both inhibit LPS-induced pathology in an in vivo endotoxemia model. Whereas the SC9 safety study was conducted in both male and female mice, the mouse endotoxemia studies were conducted in female mice only. Future studies will determine sex-specific LPS-induced pathologies, the contribution of co-morbidity factors, and their contribution to the efficacy of SC9 in endotoxemia models.

The rational design of a protein-protein interaction modulator is challenging as large flat surfaces without obvious clefts often mediate these interactions[44]. However, in this case, GTP binding induces a conformational change in Drp1 that exposes the druggable SWAG site, which has sufficient depth to accommodate a small molecule or peptide protein-protein interaction inhibitor.

In the absence of information on the mode of binding of **P110** to Drp1 and the conformation of **P110** bound within the pocket, we chose not to take the ligand-based approach for generating a pharmacophore. Instead, the protein structure-based pharmacophore approach was used. We generated an 8-point pharmacophore model with 3 donors, 3 acceptors, 1 donor or acceptor, and 1 aromatic center or pi ring center and an excluded volume to model the switch I-adjacent groove. Eight features in a pharmacophore model for virtual screening is unusually high; most studies use 4 or 5-point pharmacophores. However, we rationalized that the 8-feature pharmacophore with several donor and acceptor features could more faithfully mimic a heptapeptide, with several backbone and sidechain atoms that can function as donors and acceptors. We also determined that certain conformations of **P110** can satisfy 6 out of these 8 pharmacophore features when excluded volume criteria were ignored (to take induced fit into account). Screening for molecules that satisfied all 8 features resulted in a low hit rate. Therefore, we next screened for molecules that satisfied any 7 of the 8 features. Unsurprisingly, several hits from the screen resembled peptides and sugars with many polar groups. However, there were some drug-like molecules that we explored further.

Although small molecule inhibitors of Drp1 have been described, their specificity and therapeutic utility are uncertain. Mdivi-1 was the first described Drp1 inhibitor[52]. However, recent studies have shown that the therapeutic effect of mdivi-1 may be attributed to its inhibition of complex I[54] or mitochondrial calcium uptake[55], not specifically to Drp1 inhibition. Drpitor1a, a more recently discovered competitive GTPase inhibitor, inhibits mitochondrial fragmentation and exerts therapeutic potential in cells and a cardiac ischemia-reperfusion injury mouse model[56]. However, since Drp1 has essential functions outside of the stress response[57], the long-term safety of agents that inhibit Drp1 function indiscriminately is uncertain. Indeed, Drp1 loss-of-function variants are not tolerated in vivo; homozygous variants are embryonically lethal, and loss of function heterozygous variants can lead to severe neurological dysfunction in mice[58]. Neuron-specific and cardiac Drp1 knockout and knock-down mouse models led to progressive cell degeneration and dysfunction indicating that Drp1 is not dispensable for brain or cardiac functions[59–62]. At least 75% of Drp1 protein levels were required for normal cardiac function in mice indicating limited

spare capacity[60]. Drp1 is required for mitochondrial quality control in neurons and functions synergistically with Parkin, an important mitophagy protein[60,63]. In humans, de novo variants that inhibit function result in severe neuro-muscular diseases[64]. Furthermore, it seems Drp1 has functions outside of mitochondrial division and mitophagy including peroxisomal fission[65] and synaptic endocytosis[66]. This has spurred interest in selective protein-protein interaction inhibitors, such as **P110** and **SC9**, which blocks Drp1's pathological functions without inhibiting its physiological functions. Although **P110** is a promising potential therapeutic with a wide range of potential applications and excellent safety features[28], its pharmacokinetic properties are limiting in applications that require long-term dosing or oral bioavailability.

While both competitive and uncompetitive small molecule inhibitors of Drp1 GTPase activity have been described[56,67], the assays employed in these studies would not have discovered **SC9**. **SC9** is a potent inhibitor of pathological mitochondrial fragmentation yet it is only a weak partial inhibitor of the GTPase activity in vitro (causing only 10–25% inhibition). The published screen that identified uncompetitive compounds used low GTP concentrations (0.1 mM) and set a 30% inhibition cut-off[67]. At this GTP concentration, **SC9** does not have any inhibitory activity in vitro; supraphysiologic concentrations of GTP (1 mM) are required to detect the 10–25% inhibition with **SC9**. Yet, **SC9** has a potent therapeutic effect in cells with a nM-range IC50. This finding likely dissociates the link between GTPase inhibition in vitro and therapeutic activity in vivo.

We hypothesize that inhibition of Drp1-Fis1 protein-protein interaction, which we confirmed using proximity ligation assay, is the feature responsible for the therapeutic effect of both **P110** and **SC9**. This mechanism would explain why both compounds inhibit stress-mediated mitochondrial fission but do not affect physiological mitochondrial fission, a feature that likely contributes to the good long-term safety observed following a five-month treatment with **P110**[28]. By contrast, Mff-mediated mitochondrial fission seems to act independently of LPS or our inhibitors. This finding further supports the hypothesis that Mff mediates physiological mitochondrial fission whereas Fis1 mediates mitochondrial fragmentation associated with pathological stimuli, such as LPS. We note that the partial GTPase inhibition measurement was conducted using a recombinant Drp1 with a His-tag, which has been shown to impact Drp1 self-assembly[68]. In theory, this tag could alter the inhibitory properties of **P110** and **SC9**. However, in studies using a tag-less recombinant Drp1 and Drp1 immunoprecipitated from SHSY-5Y cells we observed a similar partial inhibition effect by **P110**[22].

The benefit of **SC9** in inhibiting LPS-induced mortality in mice (Veh. 80% vs. **SC9** 33%) was comparable to that of **P110** (Veh. 77% vs. **P110** 25%)[36]. However, SC9's pharmacological features are not yet optimal; it is relatively large (466 Daltons), is too polar to get into the brain, and has a short half-life. Therefore, current efforts are focused on identifying analogs of **SC9** with improved pharmacologic properties, including a longer half-life and improved blood brain barrier penetrance. It is not clear why **SC1** and **SC3**, which mimic at least some of the features of **P110** in vitro, do not mimic **P110** effects in cells. **SC9** has fewer rotatable bonds (7) compared to **SC1** (14) and **SC3** (10). **SC1** is not selective to Drp1 since it inhibits dynamin-2 activity. **SC3** also has low predicted solubility (0.03 mg/ml) and may aggregate. **SC1** and **SC3** could also have poor plasma membrane penetrance, off target toxicity, or lower stability in culture during 24-h incubations. Exploring these possibilities and conducting further medicinal chemistry to improve the pharmacological features of **SC9** remain outside the immediate scope of this study. Increasing blood-brain barrier penetrance of **SC9** may be required to mimic the anti-neurodegeneration effects previously achieved with **P110**.

How inhibition of excessive mitochondrial fission attenuates LPS-induced endotoxemia in vivo is not yet known. Since many cell types and tissues undergo mitochondrial fragmentation in response to

pathogen-associated molecular patterns (PAMPs), danger-associated molecular patterns (DAMPs), cytokines, and chemokines[69], it is unclear which cell types in vivo are the key mediators of the therapeutic benefit. While **SC9** mimics **P110** in blocking mitochondrial dysfunction using the recently published cardiomyocyte model[36], likely, **SC9** will also mimic **P110** in inhibiting innate immune cell activation[26]. Immune cells, such as macrophages, monocytes, NK cells, B cells, effector T cells, and astrocytes, respond robustly to danger signals and have been identified as key drivers of inflammatory disease[70]. When these cell types are activated, they fragment their mitochondria and shift to aerobic glycolysis[70]. Intriguingly, inhibition of Drp1, and other interventions that maintain oxidative phosphorylation, have been found to suppress the activation of pro-inflammatory immune cells and promote the anti-inflammatory state[70]. This agrees with the ProcartaPlex immune monitoring panel analysis that we conducted; of which the top three hits include IL-13, IL33R, and M-CSF, all of which have been implicated in innate immune cell activation and the proinflammatory response[71–73]. The concordant increase in IL-10 suggests the formation of OXPHOS-competent anti-inflammatory M2 macrophages[70,74]. This hypothesis is supported by experiments done here on RAW cell macrophages. Further experiments on innate immune cells and other cell types are needed to confirm this hypothesis. Nevertheless, our work here shows that Drp1-Fis1 inhibitors, such as **P110** and **SC9**, which preserve mitochondrial function by inhibiting excessive fission, may be important tools in the emerging field of immunometabolism. As such, **SC9** represents a class of anti-inflammatory small molecules with the potential to address mitochondrial dysfunction in a wide range of diseases.

## Methods

### Peptides, compounds, and LPS
Cyclic **P110** cargo (**cP110**; used in the GTPase and the in vitro Drp1 binding assay) and BODIPY-TMR conjugated cyclic **P110** (BDY-**cP110**, used for the in vitro Drp1 binding assay), each at 99% purity, were synthesized by CPC Scientific (San Jose, CA). **cP110** uses a glutaryl linker from the C-terminal lysine to the N-terminus of DLLPRGSK. For the **cP110**-BODIPY-TMR (BDY-**cP110**), the BODIPY was conjugated to a cysteine added at the C-terminus (DLLPRGSKC), and the peptide was cyclized with a glutaryl bridge between the C-terminal lysine and the N-terminus. Linear BODIPY-TMR-AGGDLLPRGS (uncapped, used for fluorescence polarization assay) was synthesized by AnaSpec (Fremont, CA). TAT$_{47-57}$-conjugated **P110** for in culture studies (YGRKKRRQRRRGGDLLPRGS; 95% pure, with capped ends) was synthesized by Ontores (Shanghai, China). **SC1**, **SC3**, and **SC9**, with a minimum purity of 90%, were obtained through Mcule (Palo Alto, CA) from various chemical suppliers. Chemical characterization of the compounds was performed at Enamine Ltd (Kyiv, Ukraine) and is provided below. The spectral data is available in the supplementary information file (Supplementary Fig. 4, 5 and 6) . Mdivi-1 was obtained from Sigma (M0199; St. Louis, MO). For the endotoxemia model, lipopolysaccharide (LPS) was sourced from Sigma (lot#: 0000130083; L4391; Sigma, St. Louis, MO).

**4,4'-(((((4-methyl-1,3-phenylene)bis(azanediyl))bis(carbonyl))bis(azanediyl))bis(propane-3,1-diyl))bis(morpholin-4-ium) (SC1).** [1]H NMR (600 MHz, dmso) δ 8.31 (s, 1H), 7.68 (d, $J$ = 2.3 Hz, 1H), 7.49 (s, 1H), 7.09 (dd, $J$ = 8.2, 2.2 Hz, 1H), 6.90 (d, $J$ = 8.3 Hz, 1H), 6.49 (t, $J$ = 5.7, 5.7 Hz, 1H), 5.99 (t, $J$ = 5.8, 5.8 Hz, 1H), 3.55 (t, $J$ = 4.7, 4.7 Hz, 8H), 3.11 – 3.04 (m, 4H), 2.35 – 2.30 (m, 6H), 2.30 – 2.24 (m, 6H), 2.06 (s, 3H), 1.59 – 1.52 (m, 4H). [13]C NMR (151 MHz, dmso) δ 155.79, 155.70, 139.07, 138.63, 130.32, 119.73, 112.07, 110.72, 66.65, 56.33, 56.31, 53.83, 37.86, 37.81, 27.14, 17.65. HRMS (m/z): calc. for C23 H38 N6 O4 (M + ) 462.2955, obs. 462.2951.

**(4-amino-6-((4-chlorophenyl)amino)−1,3,5-triazin-2-yl)methyl 4-oxo-4-(3-phenyl-4,5-dihydro-1H- pyrazol-1-yl)butanoate (SC3).** [1]H NMR (600 MHz, dmso) δ 9.66 (s, 1H), 7.78 (d, $J$ = 8.8 Hz, 2H), 7.75 – 7.70

(m, 2H), 7.46 – 7.42 (m, 3H), 7.29 (d, $J$ = 8.9 Hz, 2H), 7.26 – 6.96 (m, 2H), 4.82 (s, 2H), 3.89 (t, $J$ = 10.1, 10.1 Hz, 2H), 3.26 (t, $J$ = 10.1, 10.1 Hz, 2H), 2.98 (t, $J$ = 6.9, 6.9 Hz, 2H), 2.73 (t, $J$ = 6.9, 6.9 Hz, 2H). [13]C NMR (151 MHz, dmso) δ 173.03, 172.67, 169.21, 166.99, 164.41, 156.84, 139.22, 131.73, 130.66, 129.19, 128.65, 126.96, 126.14, 121.73, 65.10, 44.49, 31.80, 28.80. HRMS (m/z): calc. for C23 H22 Cl N7 O3 (M + ) 479.1473, obs. 479.1464.

**N-(6-amino-1-benzyl-2,4-dioxo-1,2,3,4-tetrahydropyrimidin-5-yl)−2-((5,7-dimethyl-[1,2,4]triazolo[1,5-a]pyrimidin-2-yl)thio)-N-methylacetamide (SC9).** [1]H NMR (500 MHz, dmso) δ 10.97 (s, 1H), 7.33 (td, $J$ = 8.0, 7.8, 1.8 Hz, 2H), 7.24 – 7.18 (m, 3H), 7.16 (d, $J$ = 7.8 Hz, 2H), 7.09 (s, 1H), 5.15 – 5.02 (m, 2H), 4.12 – 4.00 (m, 2H), 2.90 (s, 3H), 2.62 (s, 3H), 2.54 (s, 3H). [13]C NMR (151 MHz, dmso) δ 170.04, 165.89, 164.57, 159.86, 155.45, 153.39, 150.52, 146.68, 136.61, 128.98, 127.47, 126.33, 110.79, 94.37, 44.82, 35.37, 35.02, 24.85, 16.88. HRMS (m/z): calc. for C21H22N8O3S (M + ) 466.1536, obs. 466.1528.

### Essential site scanning analysis (ESSA)
ESSA was performed on PDB: 3W6P, as described in http://prody.csb.pitt.edu/tutorials/essa_tutorial/. Python 3 was used to execute python codes for ESSA. The output Protein Data Bank (3W6P_gnm_zs.pdb) file was opened in MOE and a protein surface model was generated. The surface was colored according to the $Z$-scores (reflecting the effect of interaction with a ligand) of residues using a custom Scientific Vector Language code in MOE. The minimum $Z$-score was colored yellow, the intermediate $Z$-score was set to 0.7 and was colored cyan, and the maximum $Z$-score of 2 or above was colored magenta.

### Virtual screening and molecular modeling
Molecular modeling and virtual screening were carried out using MOE Software (version 2019.01). PDB: 3W6P was loaded into MOE and the system was prepared using QuickPrep functionality in MOE at the default setting. QuickPrep functionality optimizes the hydrogen bond network and performs energy minimizations in the system. After the system was prepared, Receptor 3W6P.A, Metal 3W6P.A and Ligand 3W6P.A was kept, and all other molecules were removed. Site Finder functionality in MOE was used to calculate all the putative ligand binding sites. Dummy atoms were created at the site adjacent to Switch I (SWAG) identified by the Site Finder. All the residues within 4.5 Å from the dummy atoms were defined as pocket residues. AutoPH4 tool was used to generate the 8-feature pharmacophore with excluded volume criteria at the dummy atom site in the apo generation mode. CoCoCo library[30] was used to screen for small molecules that satisfy 7 out of 8 pharmacophore features without causing clashes with the protein.

### Filtering the hits from virtual screening
The list of small molecules obtained from the above in silico screen was prepared using the Wash functionality at the dominant protonation state at pH 7.0 and was energy-minimized, using the Energy Minimize function in MOE. The structures, Receptor 3W6P.A, Metal 3W6P.A and Ligand 3W6P.A was used for the docking analyses. The small molecule hits were docked at the dummy atom site on Receptor 3W6P.A. Triangle Matcher method was used for placement and the Rigid Receptor method was used for refinement, using London dG and GBVI/WSA dG scoring functions, respectively. Binding Score (S) was used to rank-order the molecules based on predicted affinity. The online Swiss ADME tool was used to compute the predicted physiochemical properties of the hit compounds (www.swissadme.ch). The compounds with unfavorable chemical groups flagged by Brent and PAINS filtered were removed.

### Recombinant Drp1 production
The cDNA for Drp1 isoform 3 (DNM1L) was gifted by Professor Janet Shaw (University of Utah)[75]. The gene was inserted into a pET-28a vector with an N-terminal 6xHis tag (EMD Millipore; 698643, Darmstadt, Germany) by Gibson cloning (E2611S, NEB; Ipswich, MA).

Proteins were expressed in BL21 E. coli. cultured at 37 °C to an OD600 of 1.2. Protein expression was then induced with 0.1 mM IPTG, following an overnight incubation at 37 °C. Bacteria were then lysed with BugBuster Master Mix supplemented with 20 mM imidazole, 0.4 mM PMSF, and protease inhibitor cocktail (P8849; Sigma, St. Louis, MO) and adjusted to pH 7.8. Insoluble debris was pelleted by centrifugation at 30,000 × g for 30 min and the supernatant was flowed twice through Nickle His GraviTrap Columns (11-0033-99; Cytiva, Marlborough, MA,). The column was washed with 20 mL wash buffer (HEPES 20 mM pH 7.8, 0.5 M NaCl, 20 mM imidazole) and eluted in elution buffer (HEPES 20 mM pH 7.8, 0.5 M NaCl, 500 mM imidazole, containing protease inhibitor cocktail). The buffer was exchanged with a Zeba™ Spin Desalting Column, 7 K MWCO, 10 mL (89894; Thermo Fisher, Waltham, MA) loaded with HEPES 20 mM pH 7.8, 0.25 M NaCl, 1 mM DTT, and protease inhibitor cocktail and mixed and stored in 25% glycerol. Protein concentration was estimated with nanodrop, and purity was checked with gel electrophoresis and Coomassie stain. Drp1 mutants were generated with the Agilent Quick Change II site directed mutagenesis kit (200523; Agilent, Santa Clara, CA) and were expressed in parallel under identical conditions. Protein purity and yield were comparable between mutants.

## GTPase assay
Recombinant Drp1 isoform 3 GTPase activity was measured with Sigma Malachite Green Phosphate Assay Kit (MAK307; St. Louis, MO). Drp1 was diluted to 200 nM in 100 mM KCl, 10 mM MgCl2, and 1 mM DTT and incubated with 2 μM test compounds for 30 min while rotating at room temperature. This solution was then added to 0.2 ml PCR Tubes, (22-161; Genesee Scientific, San Diego, CA) preloaded with varying concentrations of GTP (R0461; Thermo Scientific, Waltham, MA), resulting in a final concentration of 100 nM Drp1 and 1 μM compounds. Plates were then incubated for up to 1 h at 37 °C. To keep GTP concentration under 250 μM the reactions above 250 mM GTP were diluted with water and loaded into clear 96-well assay plates (3795; Corning, Corning, NY). Then, the malachite green reagent was added. Absorbance was measured at 620 nm after 30 min, using a SpectraMax fluorescence microplate reader (Molecular Devices, Sunnyvale, CA). The reaction Kcat was estimated using a phosphate standard curve, which was generated by adding known concentrations of phosphate to the same GTP-containing reaction solution and assayed in parallel on each plate. The standard accounts for non-enzymatic GTP hydrolysis. The data are expressed as production of inorganic phosphate [Pi μM]/min/[Drp1 μM] (and assay carried out at pH = 7.5).

## Fluorescence polarization assay
Recombinant Drp1 was serially diluted from 1 μM to 5 nM in 50 mM KCl, 3 mM MgCl2, 1 mM DTT, 0.01% NP40 Substitute (AAJ19628K2; Fisher Scientific, Waltham, MA), and protease inhibitor cocktail (as above). 200 μM GMPPCP (M3509; Sigma, St. Louis, MO), 200 μM GTP (R0461; ThermoFisher, Waltham, MA), or 200 μM GDP (ab146529; Abcam, Cambridge, UK.) and then 10 nM (BODIPY-TMR)-AGGDLLPRGS (AnaSpec, Fremont, CA) were mixed into each fraction and 10 μL immediately plated in a black 384-well, small volume, microplate (784076; Greiner, Monroe, NC) with 6 replicates per condition. The plate was briefly centrifuged and then the BODIPY-TMR fluorescence polarization was measured with a Tecan Infinite M1000 microplate reader (14381; Tecan, Männedorf, Switzerland), after a total of 30-min incubation at room temperature. The peptide only condition was not subtracted from the data and represents the 0 points on the linear scale x-axis.

## Inhibition of BODIPY-TMR-cP110 binding to Drp1
Recombinant Drp1 was diluted to a final concentration of 500 nM in 100 mM KCl, 10 mM MgCl2, 1 mM DTT, 0.1 mg/ml 99% bovine γ-globulins (G5009; Sigma, St. Louis, MO), and protease inhibitor cocktail (as above), and was incubated for 30 minutes with 10 μM

cP110 or compounds at room temperature with rotation. Then, 500 μM GMPPCP (as above) and 1 nM BODIPY-TMR -cP110 were added (final concentrations) for a final volume of 100 μL. Drp1 was allowed to oligomerize at room temperature for 1 h while protected from light. Samples were spun at 21,000 × g at 4 °C for 1 h. Directly after centrifugation, 3 × 10 μl of the supernatant were carefully removed and placed into a Greiner 384 well black microplate (as above, in triplicate). BODIPY-TMR fluorescence (544 ex./570 em.) was measured with a Tecan Infinite M1000 microplate reader (as above).

## Mitochondrial membrane potential and mitochondrial ROS production using TMRM and MitoSOX 96-well plate assay
For the mitochondrial membrane potential assay, clear bottom black 96-well TC plates (3904; Corning, Corning, NY) were seeded with H9c2(2-1) cells (ATCC; strain BD1X, CRL-1446;) at 8000/well and allowed to adhere overnight in 100ul DMEM/10% FBS. The cells were washed with PBS and incubated for 24hrs in DMEM/10% FBS with LPS (2 μg/ml) and test compounds. This solution was removed, and the cells were incubated with a staining solution containing Hoechst 33342 1:20,000 (H3570; Thermo Fisher, Waltham, MA) and 100 nM Tetramethylrhodamine (TMRM; T668; Thermo Fisher, Waltham, MA), in FluoroBrite DMEM/4 mM Glutamine. Plates were incubated for 25 min at 37 °C, washed 3× with 150 μL warm HBSS (14025092; Gibco, Waltham, MA), and then placed in 33 uL warm FluoroBrite DMEM/4 mM Glutamine. Fluorescence readings (top) were taken with a SpectraMax fluorescence microplate reader (Molecular Devices, Sunnyvale, CA). For mitochondrial ROS production assay, cells were prepared as above except cells were stained with Hoechst (as above) 100 nM MitoTracker Deep Red FM (ThermoFisher M22426; Waltham, MA), and 5 μM MitoSOX Red Mitochondrial Superoxide Indicator (M36008; Invitrogen, Waltham, MA). After LPS treatment, cells were washed (as above) and then plated in 100 μL FluoroBrite DMEM/4 mM Glutamine and imaged at 10× magnification in a Keyence BZ-X700 light microscope (Osaka, Japan) outfitted with a live-cell incubator. Before the experiment, a program with points pre-set at the center of each well with autofocus on the MitoTracker channel. Images were analyzed in bulk with a simple ImageJ macro (version 1.53t). The MitoTracker signal was used as a mask to extract and quantify the total MitoSOX signal (integrated density) in single cells and this was normalized to the total MitoTracker footprint. Mitochondrial ROS signal in RAW 264.7 cells (ATCC: TIB-71) and HEK-293 cells (ATCC: CRL-1573) was measured after a 6 h incubation with LPS (2 μg/mL). Mouse embryonic fibroblasts (MEF; from Dr. Hiromi Sesaki, John Hopkins University; J Cell Sci. 2013 Feb 1; 126(3): 789–802) were treated with 200 μM H2O2 (216763; Sigma, St. Louis, MO) for 3hrs. MitoSOX in HEK-293, RAW 264.7, and MEFs were measured with the SpectraMax fluorescence microplate reader and normalized to the cell number using Hoechst 33342 1:10,000.

## Live-cell mitochondrial microscopy
To quantify mitochondrial morphology and membrane potential at high magnification, 160,000 × H9c2 cells were plated on a poly-D-lysine coated MatTek 35 mm glass bottom dish (P35GC-1.5-10-C; MatTek, Ashland, MA) and allowed to adhere for 24-hours in DMEM/10% FBS. The conditions were staggered by 30 min to ensure the identical time of incubation. One by one, the dishes were washed with PBS and incubated in FluoroBrite DMEM/4 mM glutamine/10%FBS supplemented with LPS (2 μg/ml) and test compounds at 2 μM for 16 h. The media was then removed and replaced with FluoroBrite DMEM/4 mM glutamine/10%FBS supplemented with 100 nM MitoTracker Deep Red FM (as above), 100 nM Tetramethylrhodamine (as above), and Hoechst 33342 1:20,000 (as above), and incubated for 25 min at 37 °C. For the staining solution stock, 1000x dye stock mix diluted in DMSO mix was precisely aliquoted and kept at −20 °C or on dry ice under foil until just before each sequential staining media prep. Cells were washed three times in warm HBSS and then placed in warm FluoroBrite DMEM/4 mM

glutamine/10%FBS supplemented with LPS 2 μg/mL and test compounds at 2 μM. The cells were imaged in a Keyence BZ-X700 light microscope outfitted with a live-cell incubator. Multicolor 60x z-stacks were collected from random points on the plate in high resolution mode for exactly 20 min. Z-stacks were converted to full focus maximum intensity projections in Keyence BZ-analyzer and then filtered with an ImageJ macro to yield both background-subtracted and binarized images. The data were then blinded to the analyzer until all analyses were completed. To correlate mitochondrial morphology with TMRM values of single cells, cells were manually selected in ImageJ (version 1.53t) and measurements were computed in both the binarized MitoTracker channel and a TMRM channel masked by the binarized MitoTracker channel. Mitochondrial size is expressed as the 2D area per binary MitoTracker particle in the 2D maximum intensity projection. Data were expressed as the total TMRM signal (integrated density) per MitoTracker positive area per cell. All complete cells absent of mitochondrial overlap with an adjacent cell were collected from the image for analysis. Only MitoTracker objects greater than 4 pixels were used for analysis to eliminate noise.

### Immunostaining and imaging of Drp1 foci and proximity ligation assay

160,000 × H9c2 cells were plated on a poly-D-lysine coated MatTek 35 mm glass bottom dish (as above) and allowed to adhere for 24-h in DMEM/10% FBS. Cells were washed with PBS and plated in DMEM/10% FBS supplemented with LPS (2 μg/ml) and test compounds at 2 μM. After 23 h, cells were re-dosed with an 2 μM compounds and LPS and incubated for 1 h. Cells were then washed 1× with PBS and fixed with 4% formaldehyde (252549; Sigma, St. Louis, MO) in PBS for 10 min. Cells were then washed once with PBS supplemented with 22 mg/ml glycine and then permeabilized with 0.1% trition-X-100 in PBS supplemented with 22 mg/ml glycine for 10 minutes. Cells were then washed 3× with PBS. For Drp1 foci immunostaining, cells were blocked at room temperature in PBS Tween 0.1% (PBST) + 1% BSA (1 h) and then incubated overnight in a humidified chamber with 40 uL of 1:50 mouse anti-Drp1 (611113; BD Biosciences, Franklin Lakes, NJ; Lot#: 7130917) and 1:66 rabbit anti-Fis1 (10956-1-AP; Proteintech, Rosemont, IL). Cells were washed 3 × 5 min with PBST and incubated with 1:1000 secondary antibodies goat anti-Rabbit IgG (H + L) DyLight 488 (35552; Invitrogen, Waltham, MA; Lot#: TK2666875) and goat anti-Mouse IgG (H + L) DyLight 650 (84545; Invitrogen, Waltham, MA; Lot#: VK309054) in PBST + 1% BSA. Cells were washed for 3 × 5 min with PBST and then mounted with 50 uL of mounting media with DAPI (DUO82040; Sigma, St. Louis, MO). The Fis1 staining (used to focus only on Drp1 polymers that are associated with the mitochondria) was thresholded to create a binary mask to extract Drp1 foci that localize to mitochondria. The Drp1 stain was background subtracted to isolate signal from foci over the background. The total integrated density of the Drp1 foci signal per total mitochondrial area per cell was quantified by segmenting each cell manually by an observer blind to the conditions using ImageJ. The images for Fig. 5a were z-stacked since the mitochondria are not all parallel to the bottom of the cell; a full cell z-stack captures polymerized Drp1 foci on all the mitochondria in the cell. For the PLA assay, the Duolink Red kit (DUO92103; Sigma, St. Louis, MO) was used as described in the manufacture's protocol. Primary antibodies were used as above with the addition of 1:66 rabbit anti-MFF (17090-1-AP; Proteintech, Rosemont, IL). The PLA foci images were background-subtracted and thresholded to create a binary image. The total PLA foci area per cell area was quantified for individual cells manually by an observer blind to the conditions using ImageJ (version 1.53t).

### Western blot analysis to determine Drp1 levels in H9c2 and MEFs

To determine Drp1 protein levels, 100 × 20 mm Vented TC plates (3904; Corning, Corning, NY) were seeded with H9c2 at 400000/plate in 10 mL DMEM/10% FBS. The cells were then incubated for 24hrs in with LPS (2 μg/ml) and test compounds, at 2 μM, or vehicle. Protein concentrations were determined using the Pierce™ BCA Protein Assay Kit (23227, Thermo Scientific, USA). Proteins (15ug) were resuspended in Laemmli sample buffer containing 2-mercaptoethanol, loaded on SDS-PAGE, and transferred onto 0.45 μm nitrocellulose membrane. as described before (Biorad Cat#1704270, Hercules, CA). Membranes were probed with the indicated antibody and then visualized by ECL. Images acquired with Bio-Rad ChemiDoc Imagers (Hercules, CA) were analyzed with Image Lab Software (version 6.0.1) to determine relative band intensity. The antibodies used in this study were: anti-Drp1 rabbit monoclonal antibody (Cell Signaling Technology; 5391; 1:1000, Danvers, MA); anti-α-Tubulin mouse monoclonal antibody (Cell Signaling Technology; 3873; 1:1000, Danvers, MA). Secondary antibodies including anti-mouse IgG (Cat#NA931V; 1:5000) and anti-rabbit IgG (Cat#NA934V; 1:5000) were obtained from GE Healthcare (Chicago, IL). Drp1 KO MEFs were validated using rabbit anti-Drp1 monoclonal antibody (Cell Signaling Technology; 8570; 1:1000, Danvers, MA) and mouse anti-β-Actin (Cell Signaling Technology; 3700; 1:1000, Danvers, MA) using the same protocol as above. Detailed antibody validation profiles are available from the aforementioned source companies.

### High resolution mass spectrometry

The identity of **SC9** was confirmed at the Stanford University Mass Spectrometry core facility (SUMS). **The SC9** methanol solution was run on LC-ESI/MS on the Waters Acquity UPLC and Thermo Exploris 240 Orbitrap mass spectrometer (Thermo Fisher, Waltham, MA). A 15-minute gradient was used, switching from water (0.1% formic acid) to acetonitrile (0.1% formic acid). The column, a Zorbax SB-C18 2.1 × 50 mm, 2.7 u from Agilent (Santa Clara, CA), was maintained at 50 °C at a flow rate of 0.3 ml/min. Effluent from the first 2 min of the run was diverted to waste. The injection volume was 2 μL. Spectra were collected in full scan MS mode with polarity switching, Orbitrap resolution 120000, a mass range of 100–1000 Da.

### ProcartaPlex mouse immune monitoring panel

Mouse serum was collected via cardiac puncture. 0.5 M EDTA pH 8.0 coated tubes and syringes were used to prevent clotting. Serum was allowed to clot for 30 min at room temperature and then centrifuged at 1000 g for 10 min at room temp. The supernatant was collected. The 48-plex mouse ProcartaPlex immune monitoring panel was run at the Stanford Immune Monitoring Center in duplicate according to the manufacturer's instruction. Standards were used to estimate ng/μl and values were normalized to vehicle mean.

### SC9 pharmacokinetics

Pharmacokinetic plasma measurements were conducted by Keystone Bioanalytical (North Wales, PA). **SC9** was dissolved in DMSO and diluted to the working standards with ACN. The MRM transition under the ESI positive mode for **SC9** was m/z 467 > 221, internal standard (Carfentanil-d5) has an MRM transition m/z 400- > 113. The mass spectrometer was a Sciex API5500 with a Shimadzu 20 AD HPLC system. The HPLC column was a Phenomenex Kinetex 2.6 μm, XB-C18, 50 × 2.1 mm. The A mobile phase was 0.1% formic acid in water, the mobile phase B was ACN. The flow rate was 0.5 mL/min. 50 mL of mouse plasma was mixed with 50 uL of internal standard (Carf.-d5 5 μg/mL) and 0.4 mL of ASCN, mixed, and centrifuged at 14000 RPM for 5 min at 4 °C. 25 mL of the upper phase was transferred to an HPLC injection vial with 250 mL of water. For brain tissue samples, 9 parts of 50% methanol/50% water were added to 1 part of the tissue and homogenized with beads at 6500 RPM for 40 s. 50 mL of the homogenate was used for sample analysis as described above.

## Mouse housing conditions
Mice were kept on under a 12 h/12 h light/dark cycle, room temperature (~22 °C), and ~50% humidity.

## Mouse LPS-induced endotoxemia study
Female BALB/cAnNCrl mice (Strain Code 028), 7 weeks old, were purchased from Charles River (Wilmington, MA). LPS doses from 0.2 mg to 0.325 mg were tested (corresponding to 10–16.67 mg/kg based on the average weight of the group). LPS at the indicated concentration was reconstituted in sterile PBS, mixed well, and then divided equally into two tubes. To each tube, a freshly prepared **SC9** stock solution (50 mg/mL in DMSO) or plain DMSO was added. The final concentration of DMSO in the injection solution was 2.3%. The experimental conditions were blinded (marked A or B) and a different scientist administered 0.2 mL of LPS solution via intraperitoneal injection, taking note of which cages got A or B and the information was then sealed in an envelope. After 4 h, the same protocol was used to re-dose with **SC9** or DMSO (10 mg/kg) in 0.1 ml PBS (no LPS). Another scientist, blinded to which cages were A and which were B, conducted the subjective scoring. Mice were weighed and scored every 8 h for 72 h with the murine sepsis score as previously described in Shrum 2014. The respiratory distress score was taken as the sum of the respiratory rate and respiratory quality score from Shrum 2014. Mice were euthanized if they developed severe tremors.

## SC9 toxicology measurements in mice
C57Bl6/J mice (Charles River; Strain Code: 632) with $n = 6$ per condition (3 male/3 female) were dosed with **SC9** (50 mg/kg) IP and subsequently observed for 72-h. This duration was chosen since that is the typical length of our sepsis experiments. At the end of the observation period, mice were sacrificed, and blood was collected via cardiac puncture. Samples were sent to the Stanford University core animal pathology facilities for standard toxicity screening. Results included biomarkers for hematologic, hepatic, renal, and musculoskeletal damage.

## Animal welfare
All animal experiments were carried out under the protocols (APLAC-33002) approved by the Institutional Animal Care and Use Committee of Stanford University.

## Statistical analysis
All data were analyzed in Prism. For the GTPase assay, three Michaelis-Menten curves were used to estimate Km and Vmax. A one-way ANOVA with Dunnett's multiple comparisons test was used to determine statistical significance between multiple groups. A $t$-test was used to compare the two groups. The Grubbs' test was used to determine if any outliers should be removed. For murine sepsis score data, a $t$-test was used to determine the significance between the two groups at the 24-h time point. For the mouse survival analysis, the 0.3 mg LPS dose was considered, and the Log-rank (Mantel-Cox) test was used to determine significance. The Gehan-Breslow-Wilcoxon test was also automatically generated by Prism and this test concurred with the first test. For the 48-Plex mouse ProcartaPlex panel, a volcano plot without multiple test correction was used to find immune markers of interest.

## Reporting summary
Further information on research design is available in the Nature Portfolio Reporting Summary linked to this article.

## Data availability
Source data are provided with this paper. Previously published crystal structures used in this study are available in the Protein Data Bank (PDB) under accession codes 4BEJ, 3W6P and 3W6O. Any data raw and/or analyzed that support the findings of this study are available from the corresponding author upon request. Source data are provided with this paper.

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

## Acknowledgements

We thank Dr. Soichi Wakatsuki for helpful discussions. This work was supported in part by DoD ALSRP Award W81XWH-22-1-0203 and NIH Award  HL52141 to D. M-R.

## Author contributions

L.R., S.P., B.H., and D. M-R. designed the experiments. L.R., S.P., S-J.L., G.H., and B.H. performed the experiments and analyzed the results. L.R., S.P., B.H., and D. M-R. wrote and revised the manuscript.

## Competing interests

L.R., S.P., B.H., and D. M-R. are inventors on pending institutional PCT application (PCT/US2023/061879) on small molecules described in this study. The remaining authors declare no competing interests.
