## [Peer Review File · Nature Communications]

REVIEWER COMMENTS

Reviewer #1 (Remarks to the Author):

This paper describes, in part, the discovery of small molecule inhibitors of DRP1. The study over all, to a chemist, appear though and self-consistent with the arguments promoted.

In terms of the chemistry put forward, there is little to critique, but there are a number of assumptions made through the design features that should be clarified further. Firstly, modelling never demonstrates anything, it can suggest and propose, but the demonstration only arises on completion of experiments that validate the hypothesis. This is especially the case where large shifts in protein structure are being proposed. This is an area that is still in its relative infancy in terms of protein molecular modelling. Other than the suggested 'swing' in the SWAG area, is there direct experimental evidence that this actually occurs?

Arguably too much emphasis has been placed on the ability of MOE to identify druggable pockets (not just in light of the above comment), but this is off-set by the mutation studies that indicate that a loss of activity occurs. In this light the synthesis of a photoaffinity or covalent analogue followed by mass spectroscopic examination would enable greater confidence to be obtained with respect to the actual binding residues.

This reviewer feels that this work would have benefitted from a description as to how the 8-point pharmacophore was developed and what other possibilities were considered – a potentially important discussion given the complexity of inhibiting protein -protein interactions.

Overall, this is a solid body of work, but one that could benefit, and have greater impact, on the development of more appropriate compounds – it is disappointing that the introduction suggests a therapeutic role for DRP1 in neurodegenerative disease, but the analogue that is put forward is one that lacks the necessary BBB penetration.

What evidence can the authors provide that the samples SC1, SC3 and SC9 are actually the stated compounds?

Reviewer #2 (Remarks to the Author):

In the submitted manuscript “Targeting an allosteric site in dynamin-related protein 1 to inhibit FIS1-mediated mitochondrial fission”, the authors make a significant advance to the field of mitochondrial quality control by identifying a small molecule inhibitor of the mitochondrial fission mechanoenzyme DRP1 and show its utility in a mouse model of septic shock. Mitochondrial fission leads to fragmentation and dysfunction of the organelle, which occurs in many different pathological scenarios in addition to sepsis, so the impact of the current work has potential to be substantial. Current small molecule inhibitors of DRP1 suffer from off target effects and a specific inhibitor will be of great use to the community. The findings are important beyond the mitochondrial fission field as their approach of targeting a nucleotide-specific bound conformation of dynamins likely could be employed for other superfamily members. The paper is well-written and easy to follow. However, there appear to be a few missing control experiments that would allow for confidence in their conclusions. Below I summarize the study followed by specific concerns.

In Figure 1, the investigators propose that the previously identified P110 peptide targets a region adjacent to Switch 1 of the GTPase domain of DRP1 (SWAG) that they term SWAG (for Switch 1 Adjacent Groove). They show direct binding of BODIPY-P110 to this region, which is a new and important finding. Previous data indicated that P110 disrupted DRP1 interactions with FIS1 in cells and this is the first demonstration to my knowledge of target identification. Importantly the investigators identify that P110 does not bind apo-DRP1 but binds with nM affinity to nucleotide bound DRP1 by fluorescence polarization. These data support that nucleotide binding cause a conformational change that allows for peptide binding. Using the Essential Site Scanning Analysis software, they identify a putative binding site and confirm this site through mutagenesis. Thus, the peptide binds uncompetitively, which is significant because, as the authors point out, DRP1 is an essential enzyme and inhibiting it in a competitive manner would be expected to be toxic.

In Figure 2, they use the MOE software package to build a pharmacophore model that was used for virtual library screening and identified several potential DRP1 SWAG ligands. From cheminformatics analyses, three compounds (SC1, SC3, and SC9) were evaluated in Figure 3 for their ability to inhibit GTPase activity of recombinant WT and mutant DRP1s with inhibition that was statistically significant but less than P110, which is not surprising given the expectations of comparing a peptide to small molecule inhibitor at this stage of inhibitor development. These compounds were also evaluated in a sedimentation-based displacement assay with BODIPY-P110 and appear to modestly displace P110 although without a positive control these data (3j) were difficult to evaluate. Admittedly this is a difficult assay given the known self-assembly properties of DRP1, but it was unclear why the FP assay used in Fig 1d was not used to estimate an affinity.

In Figure 4, the investigators apply SC1, 3, and 9 to H9c2 cardiomyocytes that are known to undergo fission upon treatment with lipopolysaccharide (LPS). Both P110 and SC9 reverse mitochondrial fragmentation, whereas SC1 has little effect and SC3 actually increases mitochondrial fragmentation.

The different outcomes of these compounds are surprising given their similarities in all assays reported in Figure 3. The authors suggest that SC3 is toxic to cells driving further fragmentation. The consequences of LPS-induced fragmentation are decreased mitochondrial membrane potential and increased reactive oxygen species, which are shown to be reversed by P110 and SC9.

In Figure 5, the investigators present experiments aimed to determine whether the inhibited fragmentation is due to interruption of DRP1 interaction with FIS1 or MFF. This is an important question since the premise is that FIS1 drives pathological fission whereas MFF drives homeostatic fission. By fluorescence microscopy, they show LPS increases DRP1 localization to the mitochondria that appears reversed upon P110 and SC9 treatment. However, the more striking observation here (5a) is that DRP1 expression greatly increases upon LPS treatment and then decreases upon incubation with P110 and SC9. These data suggest the compounds induce degradation of DRP1 in some manner as cytoplasmic staining of non-mitochondrial DRP1 is not observed. Curiously this trend of increased then decreased signal is observed for FIS1 but not MFF in proximity ligation assays presented in 5b and 5d, respectively. I am not sure how to interpret these data but as I note below it would be important to perform Western blot analyses on these cells to evaluate that the compounds are acting solely by blocking the FIS1-DRP1 interaction and not by decreasing DRP1 levels. It is also curious that DRP1 is recruited to MFF robustly in 5d but fragmentation does not occur. Why? Shouldn't this complex be active?

In Figure 6, the authors measure an IC₅₀ of 270 nM for SC9 inhibition of recombinant DRP1 and a comparable EC₅₀ of 751nM using mitochondrial membrane potential measurements using TMRM in cells. They then test SC9 for its ability to improve LPS-induced sepsis in a mouse model and find dose-dependent response in survival, increased weight, decreased sepsis score and respiratory distress. From cytokine profiling of mouse plasma, they identify that SC9 reduces many inflammatory markers.

Major Concerns

1. Target validation with DRP1^{-/-} cells.

Despite the thorough study, the following considerations mandate that SC9 be tested in DRP1-null cells that are widely used in the field.

a. modest inhibition of GTPase activity given nM affinity is unexpected. Both P110 and SC compounds only bind after substrate binds, so they would not be expected to impact the on-rate but would impact *k_{cat}* and the off-rate substantially. But the inhibition is modest (~ decrease in activity, Fig 1k). Given the structural explanation of the Switch 1 region, one would expect more potent inhibition for nM binders.

b. The above might be explained by the presence of the e N-terminal His affinity tag on their recombinant DRP1, which is known to impact DRP1 self-assembly linked to hydrolysis (see Clinton, Bauer, and Mears 2021 doi: 10.1007/978-1-0716-0676-6_4). There appears to be less DRP1 in P110 and SC9 treated cells per Figure 5A. A Western Blot or other quantification method should be performed to confidently conclude that the decreased fission upon peptide or small molecule treatment is not just due to decreased amounts of DRP1 available within the cell.

c. Within experimental error, all 3 SC compounds have same effect GTPase activity (Fig 3k) but have entirely different effects on mitochondrial morphology and function in Figure 4. This is unexpected. For SC3 that increases fission, the authors indicate that it likely derives from toxic effects. That is reasonable, but this result indicates just how sensitive mitochondrial morphology is to off-target effects and strongly suggests that DRP1-null cells should be tested for the expected changes in mitochondrial morphology and function.

2. DRP1 levels upon LPS and drug treatment

In Figure 5a as noted above, the substantial decrease in DRP1 signal upon P110 or SC9 treatment could derive from direct binding and inhibition of DRP1 activity but is not the mechanism of action the investigators are proposing and could likely derive from off-target effects that alter DRP1 post-translational modifications etc that impact protein levels. A Western Blot should be performed to confidently conclude that the decreased fission upon peptide or small molecule treatment is not just due to decreased amounts of DRP1 available within the cell.

Minor Concerns

3. Was a GTP only control used in the hydrolysis assays Malachite Green detection solutions are commonly acid solutions, this has the potential to inflate the determined hydrolytic activity of their recombinant protein through continued non-enzymatic hydrolysis of nucleotide during the 30-minute incubation period with the Malachite Green Detection Solution (DOI: 10.1016/j.ab.2003.10.038)

4. The FP expt (Figure 1d) would be better canonically displayed with the x-axis on a log scale. The author should state in methods whether peptide alone was measured and subtracted.

5. Figure 1A – Is the reported k_{cat} in sec^{-1} or min^{-1} ? Given the reported enzyme of 100nM, I am not sure how k_{cat} is calculated from the V_{max} in panel b. Details in methods would be appreciated. The pH of the enzyme reaction should also be reported. V_{max} is typically reported in moles of product per unit time.

6. Figure 3j experimental details are confusing. The experiment is to incubate DRP1 (polymerized by addition of 200 μM GMPPCP) with 1nM BODIPY-TMRcP110 (is this what was used in Figure 1 the circular version?) and then add compounds to see if the fluorescent signal stays in pellet with DRP1 or not. cP110 inhibits the best and would be expected to displace more fluorescently labelled peptide. This is not the case. The small molecules have more in supernatant. Perhaps I am just not understanding the how the data is being normalized to % cP110. Why didn't the authors measure affinities by their FP assay used in Figure 1d? I understand the polymerization issue with DRP1 but this is overcome in 1d.

7. Nice to see mdivi-1 in Fig 3j, but it is strange that is not in the hydrolysis experiments as a control. Given the controversy in the field about whether mdivi1 inhibits DRP1 activity it would be interesting to know if the authors tested this compound in their hydrolysis assays. If so, they should report it.

8. Experimental details on how mitochondrial morphology and DRP1 colocalization was quantified should be improved. e.g. As I understand it, a 2D maximum intensity projections of DRP1 from 20 z stacks were then analysed in some manner for colocalization with the FIS1 signal (5a). If true, this could give rise to false positives from the projections. The investigators have the z-stacks and could do a colocalization analysis with those.

9. A secondary method of evaluating for direct FIS1-DRP1 binding, and subsequent disruption of this interaction, would have been beneficial and a strong addition to the work - perhaps via coimmunoprecipitation of the cells used in their PLA experiments.

10. Why is DRP1-MFF not driving mitochondrial fission upon LPS and drug treatment? Figure 5d shows by proximity ligation assay that P110 or SC9 treatment do not impair MFF ability to interact with DRP1. As this is the canonical interaction to drive mitochondrial fission, it raises the interesting question of why fragmentation does not occur? The authors should discuss this and provide their thoughts on why DRP1 levels do not decrease in 5d upon P110 and SC9 treatment as they do in 5a/b.

11. Data from 2 experiments pooled in Figures 4b-e, 5b-f. Not sure why 3 independent biological replicates were not done, although I note in these experiments >100 cells were counted typically.

12. Please clarify whether P110 is DLLPRGS or the native sequence DLLPRGT. It seems to be used interchangeably in the manuscript.

13. The use of linear and circular P110 throughout the manuscript is confusing. Do they have the same activities? Are there instances in the figure/text where P110 is indicated but cP110 is intended? Does linear P110 inhibit DRP1 activity?

14. Figure 1g arrows showing molecular rotation are non-canonical and confusing.

15. The colour scheme in 2b-2d should be explained in the figure legend typically red and blue indicate charged moieties but that is not the case here.

16. The authors should consider using grey scale in Fig 5a,b,d that will provide a better contrast than current red and cyan signal. Along these lines the authors should consider avoiding the use of red and green together in figures to make them more accessible to colour-blind individuals.

17. For future studies, the authors may want to consider using MitoNeoD with a modified dihydroethidium and tert-butyl group that does not intercalate into DNA and overcomes limitations with MitoSox (see this paper that just came out <https://doi.org/10.1038/s42255-022-00591-z>)

Reviewer #3 (Remarks to the Author):

The collective body of evidence underscores the mitochondrion as a central 'player' in the biology, physiology, and pathophysiology of many disease states, including sepsis. Vital to our ability to disentangle the complexity of mitochondrial mechanisms fundamental for homeostasis, but equally important in the pathology that manifests as death, is the need for small molecular agents to 1) perturb this system experimentally and 2) offer hope as a novel therapeutic. The authors should be commended

on their achievements in this domain specific to mitochondrial fission. The manuscript is well written, and data are compelling insofar as the specific biological mechanisms are the focus. However, the relevance to sepsis or other disease states will depend upon recapitulating the observations in more valid preclinical models. A few other moderate concerns limit the overall enthusiasm, though it is perceived that these can be addressed.

Major

I encourage the authors to temper the description of Drp1-mediated fission as a binary event dissected into either pathological (Fis1) or physiological (Mff) fission and ascribe specific components (Fis1, Mff) to one or the other. The context of time and space are critically important and insufficiently studied to support these conclusions.

Sepsis induced cardiomyopathy manifests (or perhaps identified clinically) late in the development of sepsis. Other cells and tissues are much more sensitive, and in addition, not of mesenchymal origin. Though the observation (particularly Figure 4) aligns well with studies of the kidney and lung, some parallel studies using distinct cell types (epithelial, myeloid) would strengthen the relevance of the compounds tested to the organism as a whole, and in particular, the organ dysfunction more frequently challenging the clinician (e.g., acute kidney injury). The provision of inflammatory mediators in the in vivo model begs for in vitro or ex vivo analysis of SC9 in myeloid cells.

LPS is not a model of sepsis, as the preclinical studies of anti-TNF α compounds revealed to us; such agents were beneficial in many models using LPS, but notably disappointing in the more biologically poly- and mono-microbial preclinical models of sepsis and in human participants of clinical trials. I would encourage the removal of such terminology. Endotoxemia, rather is a model more appropriately described as TLR4 pathways.

What was the rationale for restricting the studies to female Balb/c mice. In general, there are notable differences between male and female mice, with female mice, overall, exhibiting a greater 'resistance' to endotoxemia. In addition, the mice are particularly young; 7 weeks equates approximately to a teenage human. Confirmatory studies in male mice and the use of mice with comorbidities (e.g., obese, aged (>40 weeks)) are needed to support the conclusions made.

Minor

Figure 4c: What was the duration of observation?

Figure 4d: How was the competing risk of death managed for the parameters monitored over time? I perceive that the mass data of Extended Figure 7 highlight the potential issue; at 40 hours, nearly all the mice receiving 0.325 mg have died; what does that mass value really represent?

Toxicity of SC9: I do not consider body mass over 72 hours as a sufficient 'test' of toxicity, and though specific, is unlikely to be sensitive. There are several scales and/or assays specific to each organ system: Bone marrow (hematological analysis); Nervous system (cognitive, behavioral), Kidney (cystatin c, GFR), hepatic (biosynthesis). I am unable to identify the data of SC9 and MSS (Extended Data Figure 6) and draw note, like body mass, that the MSS was not developed nor validated to monitor and qualify toxicity.

REVIEWER COMMENTS

REVIEWER #1:

This paper describes, in part, the discovery of small molecule inhibitors of DRP1. The study overall, to a chemist, appear though and self-consistent with the arguments promoted.

Comment 1: In terms of the chemistry put forward, there is little to critique, but there are a number of assumptions made through the design features that should be clarified further. Firstly, modelling never demonstrates anything, it can suggest and propose, but the demonstration only arises on completion of experiments that validate the hypothesis. This is especially the case where large shifts in protein structure are being proposed. This is an area that is still in its relative infancy in terms of protein molecular modelling. Other than the suggested 'swing' in the SWAG area, is there direct experimental evidence that this actually occurs?

RESPONSE 1: We agree and have softened the language on p. 3 from 'determined' to say 'Overlaying this nucleotide-bound structure with the apo structure (PDB: 4BEJ) suggested that the endogenous P110 region swings by ~9Å...' and 'The Site Finder algorithm in Molecular Operating Environment (MOE) suggested SWAG as the third largest pocket in the GTPase domain with a positive score for propensity of ligand binding⁴⁶.'

Note that, as the reviewer commented next, our mutagenesis confirmed this prediction and it allowed us to identify SC by docking to the SWAG. Furthermore, the 'swing' was also identified by comparing crystal structures of nucleotide-bound Drp1 structures (PDB: 3W6P and 3W6O) and the APO structure (PDB: 4BEJ). Finally, such a mobile switch I GTPase mechanism is conserved in other GTPases and is well described in the literature [see reference 39; Wenger, J. et al. Functional mapping of human dynamin-1-like GTPase domain based on X-ray structure analyses. PLoS One **8**, e71835 (2013)].

Comment 2: Arguably too much emphasis has been placed on the ability of MOE to identify druggable pockets (not just in light of the above comment), but this is off set by the mutation studies that indicate that a loss of activity occurs. In this light the synthesis of a photoaffinity or covalent analogue followed by mass spectroscopic examination would enable greater confidence to be obtained with respect to the actual binding residues.

RESPONSE 2: As recognized by the reviewer, our study used mutagenesis, information from crystal structural analysis of two states of Drp1 (GTP bound and APO enzyme, literature on homologs and the demonstration that SC and P110 binding and effect are dependent on nucleotide binding to Drp1. These data together led us to conclude the presence of the SWAG pocket. In addition, we now provided a better description as to how the 8-point pharmacophore was developed and what other possibilities were considered. We modified the text to reflect this discussion.

Added text on p. 8: 'In the absence of information on the mode of binding of P110 to Drp1 and the conformation of P110 bound within the pocket, we chose not to take the ligand-based approach for generating a pharmacophore. Instead, the protein structure-based pharmacophore approach was used. We generated an 8-point pharmacophore model with 3 donors, 3 acceptors, 1 donor or acceptor and 1 aromatic center or pi ring center and an excluded volume to model the switch I adjacent groove. Eight features in a pharmacophore model for virtual screening is unusually high; most studies use 4 or 5-point pharmacophores. However, we rationalized that the 8-feature pharmacophore with several donor and acceptor features could more faithfully mimic a heptapeptide, with several backbone and sidechain atoms that can function as donors and acceptors. We also determined that certain conformations of P110 can satisfy 6 out of these 8 pharmacophore features when excluded volume criteria was ignored (to take induced fit into

account). Screening for molecules which satisfied all 8 features resulted in an extremely low hit rate. Therefore, we next screened for molecules that satisfied any 7 of the 8 features. Unsurprisingly, several hits from the screen resembled peptides and sugars with several polar groups. However, there were some drug-like molecules which we explored further'

Comment 3: Overall, this is a solid body of work, but one that could benefit, and have greater impact, on the development of more appropriate compounds – it is disappointing that the introduction suggests a therapeutic role for DRP1 in neurodegenerative disease, but the analogue that is put forward is one that lacks the necessary BBB penetration.

RESPONSE 3: Indeed, our current effort is on making a pharmacologically superior analog of SC9. However, this does not take away from the impact of the study: we identified SC9, a small molecule, that inhibits protein-protein interaction by acting on the same site as P110 and mimics the biological activity in culture and *in vivo*. We now include a clarification that better analogs are needed to advance the small molecule as a therapeutic for neurodegeneration. The focus on neurodegeneration is derived from numerous studies that show P110's effects in these models, which led us to the search for small molecule mimetics. SC9 is only a hit compound and represents an essential first step in that effort. Furthermore, P110's effects are also observed in other models that are emphasized in the text such as sepsis and ischemic injury, indications that would not require blood-brain barrier penetrance.

Added text on p. 9: 'However, SC9 pharmacological features are not yet optimal; it is relatively large (466 Daltons), has too many charges to get into brain, and has a short half-life. Therefore, current efforts are focused on identifying analogues of SC9 with better drug-like properties, including better stability and better penetrance across the blood brain barrier than SC9. ... In particular, increasing blood-brain barrier penetrance of SC9 will be required to mimic the anti-neurodegeneration effects previously achieved with P110.'

Comment 4: What evidence can the authors provide that the samples SC1, SC3 and SC9 are actually the stated compounds?

RESPONSE 4: Liquid chromatography and mass spectrometry quality control data to confirm the identity purity of SC1, SC3, and SC9 were provided by the supplier and are now included (new Extended Data Table 1 and 2 and Extended Data Figs. 4-6). In addition to this, we also conducted a high-resolution mass spectrometry of SC9 to confirm the identity of this compound (new Extended data Figure 9c).

REVIEWER #2:

In the submitted manuscript "Targeting an allosteric site in dynamin-related protein 1 to inhibit FIS1-mediated mitochondrial fission", the authors make a significant advance to the field of mitochondrial quality control by identifying a small molecule inhibitor of the mitochondrial fission mechanoenzyme DRP1 and show its utility in a mouse model of septic shock. Mitochondrial fission leads to fragmentation and dysfunction of the organelle, which occurs in many different pathological scenarios in addition to sepsis, so the impact of the current work has potential to be substantial. Current small molecule inhibitors of DRP1 suffer from off target effects and a specific inhibitor will be of great use to the community. The findings are important beyond the mitochondrial fission field as their approach of targeting a nucleotide-specific bound conformation of dynamins likely could be employed for other superfamily members. The paper is well-written and easy to follow. However, there appear to be a few missing control experiments that would allow for confidence in their conclusions. Below I summarize the study followed by specific concerns.

In Figure 1, the investigators propose that the previously identified P110 peptide targets a region

adjacent to Switch 1 of the GTPase domain of DRP1 (SWAG) that they term SWAG (for Switch 1 Adjacent Groove). They show direct binding of BODIPY-P110 to this region, which is a new and important finding. Previous data indicated that P110 disrupted DRP1 interactions with FIS1 in cells and this is the first demonstration to my knowledge of target identification. Importantly the investigators identify that P110 does not bind apo-DRP1 but binds with nM affinity to nucleotide bound DRP1 by fluorescence polarization. These data support that nucleotide binding cause a conformational change that allows for peptide binding. Using the Essential Site Scanning Analysis software, they identify a putative binding site and confirm this site through mutagenesis. Thus, the peptide binds uncompetitively, which is significant because, as the authors point out, DRP1 is an essential enzyme and inhibiting it in a competitive manner would be expected to be toxic.

Comment 5: In Figure 2, they use the MOE software package to build a pharmacophore model that was used for virtual library screening and identified several potential DRP1 SWAG ligands. From cheminformatics analyses, three compounds (SC1, SC3, and SC9) were evaluated in Figure 3 for their ability to inhibit GTPase activity of recombinant WT and mutant DRP1s with inhibition that was statistically significant but less than P110, which is not surprising given the expectations of comparing a peptide to small molecule inhibitor at this stage of inhibitor development. These compounds were also evaluated in a sedimentation-based displacement assay with BODIPY-P110 and appear to modestly displace P110 although without a positive control these data (3j) were difficult to evaluate. Admittedly this is a difficult assay given the known self-assembly properties of DRP1, but it was unclear why the FP assay used in Fig 1d was not used to estimate an affinity.

RESPONSE 5: In 3j, controls were included: unlabeled cP110 was used as a positive control to displace labeled cP110 and mdivi-1, which binds Drp1 at a different site than P110, was used as a negative control. We now emphasize that in the text.

We used GTPase assay to estimate the affinity of the compound because this is the standard practice and because the GTPase assay provided a higher signal to noise ratio than the FP and sedimentation assays, making this a better method to estimate K_d.

Edited text on p. 4: 'Pre-treating with a large excess of unlabeled cP110 (10μM) blocks all P110-specific BDY-cP110 binding to Drp1, resulting in an increased BDY signal in the supernatant (Fig. 3j, cP110) and acts as a positive control in our assay. All three compounds also reduced the co-sedimentation of BDY-cP110 with recombinant Drp1, whereas Mdivi-1⁵⁰, another Drp1 inhibitor, did not (Fig. 3j) and thus serving as a negative control in this assay.'

Comment 6: In Figure 4, the investigators apply SC1, 3, and 9 to H9c2 cardiomyocytes that are known to undergo fission upon treatment with lipopolysaccharide (LPS). Both P110 and SC9 reverse mitochondrial fragmentation, whereas SC1 has little effect and SC3 actually increases mitochondrial fragmentation. The different outcomes of these compounds are surprising given their similarities in all assays reported in Figure 3. The authors suggest that SC3 is toxic to cells driving further fragmentation. The consequences of LPS-induced fragmentation are decreased mitochondrial membrane potential and increased reactive oxygen species, which are shown to be reversed by P110 and SC9.

RESPONSE 6: *In silico* hit compound activity in a recombinant biochemical assay do not always translate to activity in cell-based assays; cell-based assays introduce confounding variables, such as the ability to cross the cell membrane, off-target toxicity, stability in serum during 24-hour incubations at 37°C, and many other variables. The cell-based assays were used for this reason since it can screen out compounds that are less suitable for further biological testing. We added new text to address the reviewer's concern.

Edited text on p. 9: 'It is not clear why SC1 and SC3 mimic P110 *in vitro* but not in cells. SC9 has fewer rotatable bonds (7) compared to SC1 (14) and SC3 (10). SC1 is not selective to Drp1 since it inhibits dynamin-2 activity. SC3 also has low predicted solubility (0.03mg/ml) and may aggregate. SC1 and SC3 could also have poor plasma membrane penetrance, off target toxicity, or lower stability in culture

during 24-hour incubations. Exploring these possibilities and conducting further medicinal chemistry to improve the pharmacological features of SC9 remain outside the immediate scope of this study.'

Comment 7: In Figure 5, the investigators present experiments aimed to determine whether the inhibited fragmentation is due to interruption of DRP1 interaction with FIS1 or MFF. This is an important question since the premise is that FIS1 drives pathological fission whereas MFF drives homeostatic fission. By fluorescence microscopy, they show LPS increases DRP1 localization to the mitochondria that appears reversed upon P110 and SC9 treatment. However, the more striking observation here (5a) is that DRP1 expression greatly increases upon LPS treatment and then decreases upon incubation with P110 and SC9. These data suggest the compounds induce degradation of DRP1 in some manner as cytoplasmic staining of non-mitochondrial DRP1 is not observed. Curiously this trend of increased then decreased signal is observed for FIS1 but not MFF in proximity ligation assays presented in 5b and 5d, respectively. I am not sure how to interpret these data but as I note below it would be important to perform Western blot analyses on these cells to evaluate that the compounds are acting solely by blocking the FIS1-DRP1 interaction and not by decreasing DRP1 levels. It is also curious that DRP1 is recruited to MFF robustly in 5d but fragmentation does not occur. Why? Shouldn't this complex be active?

RESPONSE 7: The Drp1 foci imaging assay does not detect all the cellular Drp1; it only measures the abundance of large Drp1 polymers, the Drp1 species that is seen as foci in the image (Fig. 5a; Extended Data Fig. 8a). Furthermore, we restricted the analysis to only Drp1 foci that lay within the mitochondrial footprint. The lower order Drp1 oligomers such as monomers, dimers, and tetramers do not appear as foci and are therefore not captured in this assay. We have clarified this now in the text. In addition, we have included a western blot that directly measures total Drp1 to show that the levels of Drp1 do not change with LPS, P110, or SC9 treatments; (see new Figure 5c and Extended Data fig. 8c; shown below).

Added text on p. 5: 'Using immunocytochemistry, we first determined the number of large oligomeric Drp1 species on the mitochondria in single cells. Large Drp1 oligomers, which are the active species of Drp1¹⁸, appear as foci in this assay, and their number and size can be quantified by the fluorescence signal (Extended Data Fig. 8a). This method does not measure total Drp1 levels, since lower order monomers, dimer, and tetramers are not detected. Furthermore, we restricted our analysis to detects only polymerized Drp1 (the Drp1 state that mediates mitochondrial fission) on the surface of the mitochondria. (The images for Fig. 5a were z-stacked, since the mitochondria are not all parallel to the bottom of the cell; a full cell z-stack captures polymerized Drp1 foci on all the mitochondria in the cell.) LPS increased the amount of mitochondria-associated Drp1 foci by about two-fold and both P110 and SC9 reduced the signal to below the level of control (Fig. 5a,b; Extended Data Fig. 8a,b). To confirm that the increase in Drp1 oligomers was not due to a change in

the total Drp1 protein levels, we conducted western blots on whole cell lysates and observed no difference in total Drp1 protein levels after treatment (Fig. 5c; Extended Data Fig. 8c). This indicates that P110 and SC9 block stress-induced Drp1 polymerization, a process that occurs on the mitochondrial surface.'

In Figure 6, the authors measure an IC50 of 270 nM for SC9 inhibition of recombinant DRP1 and a comparable EC50 of 751nM using mitochondrial membrane potential measurements using TMRM in cells. They then test SC9 for its ability to improve LPS-induced sepsis in a mouse model and find dose-dependent response in survival, increased weight, decreased sepsis score and respiratory distress. From cytokine profiling of mouse plasma, they identify that SC9 reduces many inflammatory markers.

Major Concerns

Comment 8: 1. Target validation with DRP1^{-/-} cells. Despite the thorough study, the following considerations mandate that SC9 be tested in DRP1-null cells that are widely used in the field.

RESPONSE 8: The reviewer's suggestion is aimed to examine whether SC9's benefit is dependent on Drp1. If SC9's benefit is via Drp1 engagement– SC9 is expected to have no effect in Drp1 KO cells. We conducted the experiment suggested by the reviewer. Previously (<https://www.ncbi.nlm.nih.gov/pmc/articles/PMC7497224/pdf/IMCB-98-528.pdf>), it was shown that LPS doesn't induce mitochondrial fragmentation in Drp1 KO MEFs. We therefore switched to H2O2 as the stressor, as we have done before (Joshi Nature Neurosci ref). We used 200 uM H2O2, which was determined to be the maximum tolerated dose above which we observed significant necrotic cell death (LDH release assay; data not shown). In WT MEFs, we found a 2-fold increase in mitoROS that was blocked by SC9. However, consistent with results with LPS, Drp1 KO cells did not exhibit mitochondria-dependent response i.e. elevation of mitoROS. However, as expected, SC9 also did not affect basal Mito ROS levels (See Figure). The lack of mitochondria-dependent increase in ROS production is also expected; Drp1-dependent excessive fission is needed to occur to produce mitochondrial-dependent ROS production and in Drp1 KO – this cannot occur. With these data taken together, we conclude that since cells lacking Drp1 do not respond to SC9, SC9 effect is Drp1-dependent. We thank the reviewer for this suggestion.

Comment 9: a. modest inhibition of GTPase activity given nM affinity is unexpected. Both P110 and SC compounds only bind after substrate binds, so they would not be expected to impact the on-rate but would impact kcat and the off-rate substantially. But the inhibition is modest (~ decrease in activity, Fig 1k). Given the structural explanation of the Switch 1 region, one would expect more potent inhibition for nM binders

RESPONSE 9: The partial GTPase inhibitory effect of P110 is not a new finding; we previously reported this in both Kornfeld *et al.*, Sci Rep. 2018;8:14034 (see figure 1h from that study, attached on the right) and Qi *et al.*, J Cell Sci. 2013;126:789-802. The reason why these compounds are partial inhibitors may be attributed to their uncompetitive allosteric inhibitory effect (not competitive for GTP binding site). This can have complex effects on the GTPase-induced Drp1 aggregation and on Drp1-mediated excessive mitochondrial fission. We also demonstrated that P110's effect in cells is due to inhibition of Drp1-Fis1 protein-protein interaction (*ibid*). Since SC9 has even less GTPase inhibitory effect on Drp1, yet has a potent effect on inhibiting stress-induced mitochondrial dysfunction to

(Fig. 1h in Kornfeld et al., 2018)

comparable or even greater extent than P110, we believe this is evidence to support the original claim that the mechanism by which P110, and now SC9, act through Drp1-Fis1 protein-protein interaction inhibition in cells.

Comment 10: b. The above might be explained by the presence of the N-terminal His affinity tag on their recombinant DRP1, which is known to impact DRP1 self-assembly linked to hydrolysis (see Clinton, Bauer, and Mears 2021 doi: 10.1007/978-1-0716-0676-6_4).

RESPONSE 10: We agree with the reviewer and this caveat was added. However, we previously demonstrated a partial inhibition of tag-less Drp1 by P110 and a similar partial inhibition of immunoprecipitated Drp1 from SHSY-5Y cells (Qi *et al.*, *ibid*).

Added text on p. 9: ‘We note that the partial GTPase inhibition measurement was conducted using a recombinant Drp1 with a His-tag, which has been shown to impact Drp1 self-assembly⁶⁶. In theory, this tag could alter the inhibitory properties of P110 and SC9. However, in previous studies using a tag-less recombinant Drp1 and Drp1 immunoprecipitated from SHSY-5Y cells we observed a similar partial inhibition effect by P110.’

Comment 11: There appears to be less DRP1 in P110 and SC9 treated cells per Figure 5A. A Western Blot or other quantification method should be performed to confidently conclude that the decreased fission upon peptide or small molecule treatment is not just due to decreased amounts of DRP1 available within the cell.

RESPONSE 11: See response 7, above.

Comment 12: a. Within experimental error, all 3 SC compounds have same effect GTPase activity (Fig 3k) but have entirely different effects on mitochondrial morphology and function in Figure 4. This is unexpected. For SC3 that increases fission, the authors indicate that it likely derives from toxic effects. That is reasonable, but this result indicates just how sensitive mitochondrial morphology is to off-target effects and strongly suggests that DRP1-null cells should be tested for the expected changes in mitochondrial morphology and function.

RESPONSE 12: See response 6, above. Note also that SC9 had no effect on naive cells and control mice, *in vivo* (Extended Data Fig. 9), suggesting that SC9’s inhibition of mitochondrial fission only in the presence of LPS is specific.

Comment 13: 2. DRP1 levels upon LPS and drug treatment

In Figure 5a as noted above, the substantial decrease in DRP1 signal upon P110 or SC9 treatment could derive from direct binding and inhibition of DRP1 activity but is not the mechanism of action the investigators are proposing and could likely derive from off-target effects that alter DRP1 post-translational modifications etc. that impact protein levels. A Western Blot should be performed to confidently conclude that the decreased fission upon peptide or small molecule treatment is not just due to decreased amounts of DRP1 available within the cell.

RESPONSE 13: See response 7, above.

Minor Concerns:

Comment 14: 3. Was a GTP only control used in the hydrolysis assays? Malachite Green detection solutions are commonly acid solutions, this has the potential to inflate the determined hydrolytic activity of their recombinant protein through continued non-enzymatic hydrolysis of nucleotide during the 30-minute incubation period with the Malachite Green Detection Solution (DOI: 10.1016/j.ab.2003.10.038)

RESPONSE 14: A GTP only control (all reagents, but no Drp1) was used to create the phosphate standard. To create this standard, known concentrations of phosphate were spiked into this solution and assayed in parallel on the same plate to estimate the amount of phosphate generated by Drp1

specifically. This controls for any potential artifacts including due to non-enzymatic GTP hydrolysis. This is now clarified in Methods.

Edited text on p. 18: 'The reaction Kcat was estimated using a phosphate standard curve, which was generated by adding known concentrations of phosphate to the same GTP-containing reaction solution and assayed in parallel on each plate. The standard accounts for non-enzymatic GTP hydrolysis.'

Comment 15: 4. The FP expt (Figure 1d) would be better canonically displayed with the x-axis on a log scale. The author should state in methods whether peptide alone was measured and subtracted.

RESPONSE 15: The linear axis was used because this allows for the display of the peptide only data (see values for 0 Drp1), whereas the log axis could not include this data point. This is now emphasized in the Method section.

Added text on p. 19: 'The peptide only condition was not subtracted from the data and represents the 0 point on the linear scale axis.'

Comment 16: 5. Figure 1A – Is the reported kcat in sec⁻¹ or min⁻¹? Given the reported enzyme of 100nM, I am not sure how kcat is calculated from the Vmax in panel b. Details in methods would be appreciated. The pH of the enzyme reaction should also be reported. Vmax is typically reported in moles of product per unit time.

RESPONSE 16: The legend and method were updated according to the reviewer's suggestion to indicate that the Kcat and Vmax are reported as [Pi mM]/min/[Drp1 mM] (Fig. 1 a-c). The pH of the assay is also now indicated.

Added text on p. 18: 'The reaction Kcat was estimated using a phosphate standard curve, The data are expressed as production of inorganic phosphate [Pi μM]/min/[Drp1 μM] (and assay carried out at pH=7.5).

Comment 17: 6. Figure 3j experimental details are confusing. The experiment is to incubate DRP1 (polymerized by addition of 200 μM GMPPCP) with 1nM BODIPY-TMRcP110 (is this what was used in Figure 1 the circular version?) and then add compounds to see if the fluorescent signal stays in pellet with DRP1 or not. cP110 inhibits the best and would be expected to displace more fluorescently labelled peptide. This is not the case. The small molecules have more in supernatant. Perhaps I am just not understanding the how the data is being normalized to % cP110.

RESPONSE 17: The reviewer describes the data correctly. However, since there is no statistical difference between the effect of cP110 and SC compounds in Fig. 3j, the interpretation that the compounds result in greater signal in supernatant than P110 is incorrect.

Comment 18: Why didn't the authors measure affinities by their FP assay used in Figure 1d? I understand the polymerization issue with DRP1 but this is overcome in 1d.

RESPONSE 18: The GTPase assay was used to estimate the affinity of the compound, since this is a standard practice for enzymes. Importantly, the GTPase assay provided a greater signal to noise than the FP and sedimentation assays, making this method superior to estimate Kd.

Comment 19: 7. Nice to see mdivi-1 in Fig 3j, but it is strange that is not in the hydrolysis experiments as a control. Given the controversy in the field about whether mdivi1 inhibits DRP1 activity it would be interesting to know if the authors tested this compound in their hydrolysis assays. If so, they should report it.

RESPONSE 19: Unfortunately, we have not included Mdivi-1 in our GTPase assays, as its effect *in vitro* [reported by Nunnari (Dev Cell. 2008;14:193-204)] is not in dispute as far as we know; the debated effects of Mdivi-1 relate to the biological activity of the compound.

Comment 20: 8. Experimental details on how mitochondrial morphology and DRP1 colocalization was quantified should be improved. e.g., As I understand it, a 2D maximum intensity projections of DRP1 from 20 z stacks were then analyzed in some manner for colocalization with the FIS1 signal (5a). If true,

this could give rise to false positives from the projections. The investigators have the z-stacks and could do a coloc2 analysis with those.

RESPONSE 20: We clarified the text (page 9) and the experimental details (see p. 19-20 and below). We now emphasize that in figure 5a, the assay detects only polymerized Drp1 (the Drp1 state that mediates mitochondrial fission) on the surface of the mitochondria and a direct measure of Drp1-Fis1 colocalization was demonstrated by the proximity ligation assay (PLA; Fig. 5d-f).

The images for Fig. 5a were z-stacked, since the mitochondria are not all parallel to the bottom of the cell; a full cell z-stack captures polymerized Drp1 foci on all the mitochondria in the cell. Furthermore, Fis1, which was used to label the mitochondria, is distributed throughout the mitochondrial surface (green in Extended Data 8a), suggesting that only a small amount of Fis1 is occupied with Drp1 polymers. Therefore, coloc2 analysis is not appropriate. Finally, since not all the polymerized Drp1 are necessarily attached to the mitochondria, the analysis of Drp1 foci was restricted to the Fis1+ mitochondrial signal. We used PLA as the appropriate assay to determine Drp1-Fis1 co-localization; it captures distances shorter than 40nm and would not have the same false positive issue from z-stacking.

Modified text on p. 5: 'This method does not measure total Drp1 levels, since lower order monomers, dimer, and tetramers are not detected. Furthermore, we restricted our analysis to detects only polymerized Drp1 (the Drp1 state that mediates mitochondrial fission) on the surface of the mitochondria. (The images for Fig. 5a were z-stacked, since the mitochondria are not all parallel to the bottom of the cell; a full cell z-stack captures polymerized Drp1 foci on all the mitochondria in the cell.)'

Modified text on p. 20: 'Mitochondrial size is expressed as the 2D area per binary MitoTracker particle in the 2D maximum intensity projection. Data were expressed as the total TMRM signal (integrated density) per MitoTracker positive area per cell. All complete cells absent of mitochondrial overlap with an adjacent cell were collected from the image for analysis. Only MitoTracker objects greater than 4 pixels were used for analysis to eliminate noise.'

Modified text on p. 20: 'The Fis1 staining (used to focus only on Drp1 polymers that are associated with the mitochondria) was thresholded to create a binary mask to extract Drp1 foci that localize to mitochondria. The Drp1 stain was background subtracted to isolate signal from foci over background. The total integrated density of the Drp1 foci signal per total mitochondrial area per cell was quantified by segmenting each cell manually by an observer blind to the conditions using ImageJ.'

Comment 21: 9. A secondary method of evaluating for direct FIS1-DRP1 binding, and subsequent disruption of this interaction, would have been beneficial and a strong addition to the work - perhaps via coimmunoprecipitation of the cells used in their PLA experiments.

RESPONSE 21: Indeed, a co-IP of Drp1-Fis1 interaction could be used and has been shown for P110 in our previous studies (Qi *et al.*, J Cell Sci. 2013;126:789-802). However, in our experience, the PLA method is superior, since co-IP averages the effect of all cells and these data can be affected by dead cells and debris.

Comment 22: 10. Why is DRP1-MFF not driving mitochondrial fission upon LPS and drug treatment? Figure 5d shows by proximity ligation assay that P110 or SC9 treatment do not impair MFF ability to interact with DRP1. As this is the canonical interaction to drive mitochondrial fission, it raises the interesting question of why fragmentation does not occur? The authors should discuss this and provide their thoughts on why DRP1 levels do not decrease in 5d upon P110 and SC9 treatment as they do in 5a/b.

RESPONSE 22: This is a central point of our quest to generate a selective inhibitor of Drp1-Fis1 interaction, an interaction that occurs during pathological stimuli. P110 was rationally designed to be a selective Drp1-Fis1 interaction inhibitor (Qi *et al.*, J Cell Sci. 2013;126:789-802). Our work here

replicates these findings with PLA and shows that SC9 is mimicking P110's Fis1 selectivity. Mff mediates mitochondrial physiological fission, however it does so independently of LPS or our inhibitors (Fig. 5e). See also differential regulation of Fis1 and Mff-mediated fission in other publications (e.g., Kleele *et al.*, 2021. Distinct fission signatures predict mitochondrial degradation or biogenesis. *Nature* 593, 435–439.) Simply stated, basal fission that occurs via Mff/Drp1 are unaffected by SC9 and P110, nor by LPS, hence the lack of effect on PLA. We thank you for helping us clarify this important point. Also see response 31 below.

Modified text on p.9: 'We hypothesize that inhibition of Drp1-Fis1 protein-protein interaction, which we confirmed using proximity ligation assay, is the feature responsible for the therapeutic effect of both P110 and SC9. This mechanism would explain why both compounds inhibit stress-mediated mitochondrial fission but do not affect physiological mitochondrial fission, a feature that likely contributes to the good long-term safety observed following a five-month treatment with P110²⁸. By contrast, Mff-mediated mitochondrial fission seems to act independently of LPS or our inhibitors. This finding further supports the hypothesis that Mff mediates physiological mitochondrial fission whereas Fis1 mediates mitochondrial fragmentation associated with pathological stimuli, such as LPS.'

Comment 23: 11. Data from 2 experiments pooled in Figures 4b-e,5b-f. Not sure why 3 independent biological replicates were not done, although I note in these experiments >100 cells were counted typically.

RESPONSE 23: Indeed, the two experiments obtained over 200 single cell measurements, sufficiently powered to provide comparison of the conditions; each data point represents a single cell and all our measures (membrane potential, ROS, Fis1-Drp1 interaction, and Drp1 polymerization) are known to be covariates. The sum of all these measurements supports the finding the LPS induces stress and SC9 and P110 block this response.

Comment 24: 12. Please clarify whether P110 is DLLPRGS or the native sequence DLLPRGT. It seems to be used interchangeably in the manuscript.

RESPONSE 24: DLLPRGT is referred to as the "endogenous P110 homology sequence." The P110 peptide was rationally designed from this sequence and the Fis1 sequence and is DLLPRGS. This is now clarified in the text.

Modified text on p. 2: 'The P110 heptapeptide (DLLPRGS) was rationally designed from such a short homology sequence between hDrp1 (49-55; DLLPRGT) and hFis1 (60-66; ELLPKGS)²². Cross-linking of the P110 heptapeptide to TAT₄₇₋₅₇ enabled peptide delivery into cells in culture and *in vivo*. The term P110 hereafter refers to the TAT conjugated P110 heptapeptide.'

Comment 25: 13. The use of linear and circular P110 throughout the manuscript is confusing. Do they have the same activities? Are there instances in the figure/text where P110 is indicated but cP110 is intended? Does linear P110 inhibit DRP1 activity?

RESPONSE 25: Linear P110 is a TAT-conjugated peptide while cP110 does not have TAT and therefore must be made cyclic for stability reasons. The heptapeptide that constitutes the active binding agent in P110 is referred to as "the P110 heptapeptide." We have now clarified this in the text (see RESPONSE 24). We found that cP110-TAT also has a higher stability in cells, but its biological activity is the same as the original P110-TAT. This work was previously conducted and is not included in this manuscript.

Comment 26: 14. Figure 1g arrows showing molecular rotation are non-canonical and confusing.

RESPONSE 26: We removed these arrows (see edited Fig. 1g.)

Comment 27: 15. The colour scheme in 2b-2d should be explained in the figure legend typically red and blue indicate charged moieties but that is not the case here.

RESPONSE 27: While this is common for many applications, in chemical structures red is commonly used to denote oxygen and blue for nitrogen. This is now clarified in the legend of Fig. 2.

Comment 28: 16. The authors should consider using grey scale in Fig 5a,b,d that will provide a better contrast than current red and cyan signal. Along these lines the authors should consider avoiding the use of red and green together in figures to make them more accessible to colour-blind individuals.

RESPONSE 28: Since the images in 4a, 5b, and 5d show two channels, the use of color allows the reader to differentiate the nuclear signal (blue) from the experimental signal. The figures were run through a color blind simulator to ensure that all information can be extracted by color blind individuals.

Comment 29: 17. For future studies, the authors may want to consider using MitoNeoD with a modified dihydroethidium and tert-butyl group that does not intercalate into DNA and overcomes limitations with MitoSox (see this paper that just came out <https://doi.org/10.1038/s42255-022-00591-z>)

RESPONSE 29: We thank the reviewer for letting us know of this dye. In this study, microscopy and image analysis were used to extract the cytoplasmic signal and remove the nuclear signal. The results should still be valid here though the improved reagent is noted.

Blake Hill

REVIEWER #3:

Comment 30: The collective body of evidence underscores the mitochondrion as a central ‘player’ in the biology, physiology, and pathophysiology of many disease states, including sepsis. Vital to our ability to disentangle the complexity of mitochondrial mechanisms fundamental for homeostasis, but equally important in the pathology that manifests as death, is the need for small molecular agents to 1) perturb this system experimentally and 2) offer hope as a novel therapeutic. The authors should be commended on their achievements in this domain specific to mitochondrial fission. The manuscript is well written, and data are compelling insofar as the specific biological mechanisms are the focus. However, the relevance to sepsis or other disease states will depend upon recapitulating the observations in more valid preclinical models.

RESPONSE 30: We agree with the reviewer that the small molecule inhibitor offers promise as a novel therapeutic and that the LPS model of sepsis does not model human clinical sepsis well. To address this comment, we have removed the use of the term sepsis and replaced it with endotoxemia. The core claim in this paper is that SC9 is a P110 mimetic. The statements around the general translational potential of SC9 are derived from the fact that P110 has been shown to have promise in many clinical models, including Huntington’s disease (HD)²²⁻²⁶, amyotrophic lateral sclerosis (ALS)^{25,27}, Alzheimer’s disease (AD)^{25,28}, Parkinson’s disease (PD)^{21,29,30}, scrapie³¹, cardiac ischemia³², stroke³³, inflammatory bowel disease³⁴, and endotoxemia^{35,36}, in mouse and patient-derived cell models. What has been limiting P110’s translation is not the pre-clinical animal models but the pharmacokinetic limitations of P110 as a peptide. The discovery that a small molecule can mimic P110 opens new pharmacokinetic opportunities that peptides struggle to achieve. Translating these small molecules into the clinic will require much more development and validation including improved blood brain barrier penetration. However, we believe that SC9 is an essential step in that effort.

A few other moderate concerns limit the overall enthusiasm, though it is perceived that these can be addressed.

Major

Comment 31: I encourage the authors to temper the description of Drp1-mediated fission as a binary event dissected into either pathological (Fis1) or physiological (Mff) fission and ascribe specific components (Fis1, Mff) to one or the other. The context of time and space are critically important and insufficiently studied to support these conclusions.

RESPONSE 31: We agree with the reviewer that the biology of Fis1 and Mff in cells and *in vivo* is more complex. Our claim is more specific than the proteins themselves. The claim is that Fis1-Drp1 interaction (not the Fis1 protein) is specific to cell stress. Indeed, Fis1 has other functions through interactions with other proteins. Pharmacologic protein-protein interaction inhibitors indicate that Fis1-Drp1 interaction has no effect under basal physiological conditions yet a profound effect in numerous models of pathological cell stress. In contrast, Drp1-Mff disruption pharmacologically or genetically is toxic under basal conditions, indicating that Mff performs essential functions. Mff may also contribute to pathological fission, however as we have shown here (Fig. 5g) and before (Qi *et al.*, *J Cell Sci.* 2013;126:789-802; Kornfeld *et al.*, *Sci Rep.* 2018;8:14034) Drp1-Mff interaction does not increase during cell stress and Mff inhibition is not a viable strategy of inhibiting cell stress since this is toxic. We thank the reviewer for pointing out the lack of clarity on this issue and we have therefore added the following.

Modified text on p. 2: 'Importantly, P110 has no apparent toxicity in mice even after five months of sustained delivery at 5mg/kg/day²⁸. Fis1 may also have physiological roles through its interaction with other proteins; however, Drp1-Fis1 interaction is specifically induced during inflammatory cell death^{22,26,39}. Fis1 knock-down with siRNA does not have an impact on basal mitochondrial fission⁴⁰, and Fis1-KO has a minor phenotype⁴¹. In contrast, Drp1-Mff interaction is found in both basal and stress conditions and Mff-KO has a profound effect on basal mitochondrial fission resulting in severely elongated mitochondria⁴¹. Mff may also recruit Drp1 during cell death and some genetic models indicate that Mff knock-down can reduce cell death in culture and infarct size during ischemia-reperfusion injury^{41,42}. However, inhibition of Mff is not a viable strategy to reduce inflammatory cell death since pharmacologic and genetic perturbation of Mff is toxic in several cell-types and *in vivo*^{20,21}. This is because Mff-Drp1 interaction governs essential functions under basal conditions whereas Fis1-Drp1 interaction does not.'

Comment 32: Sepsis induced cardiomyopathy manifests (or perhaps identified clinically) late in the development of sepsis. Other cells and tissues are much more sensitive, and in addition, not of mesenchymal origin. Though the observation (particularly Figure 4) aligns well with studies of the kidney and lung, some parallel studies using distinct cell types (epithelial, myeloid) would strengthen the relevance of the compounds tested to the organism as a whole, and in particular, the organ dysfunction more frequently challenging the clinician (e.g., acute kidney injury). The provision of inflammatory mediators in the *in vivo* model begs for *in vitro* or *ex vivo* analysis of SC9 in myeloid cells.

RESPONSE 32: The H9c2 cell experiments were not necessarily used to make claims about the cellular mechanisms of disease pathology *in vivo*. This cell model was used to determine if SC9 is mimicking P110 in cells. We have previously demonstrated the efficacy of P110 on endotoxemia-induced multi-organ dysfunction syndrome (MODS) in cardiac injury (Haileselassie, B. *et al.*, 2019; Drp1/Fis1 interaction mediates mitochondrial dysfunction in septic cardiomyopathy. *J Mol Cell Cardiol* **130**, 160–169), neuronal injury (Haileselassie, B. *et al.* 2020; Mitochondrial dysfunction mediated through dynamin-related protein 1 (Drp1) propagates impairment in blood brain barrier in septic encephalopathy. *J Neuroinflammation* **17**, 36), and macrophage dysfunction (Lin *et al.*, 2022; δ PKC-mediated Drp1 phosphorylation impacts macrophage mitochondrial function and inflammatory response to endotoxin. *Shock* 57:435-443; Mukherjee *et al.*, 2022; Drp1/Fis1-dependent pathologic fission and associated damaged extracellular mitochondria contribute to macrophage dysfunction in endotoxin tolerance. *Crit Care Med.* 50:e504-e515), suggesting that Fis1-Drp1 interaction is an important mechanism in the pathobiology of endotoxemia. To address the

reviewer's comment and further validate SC9, we measured the efficacy of SC9 on LPS-mediated mitochondrial injury using HEK (kidney) and RAW cells (macrophages) and added these new data.

Modified text on p. 7: 'The disease pathophysiology of LPS-induced endotoxemia *in vivo* involves many cell types. To determine if SC9 inhibits LPS-induced mitochondrial dysfunction in other cell-types, we repeated the assays in two other cell types. Like cardiac H9c2 cells, HEK293 cells (human embryonic kidney cells) and RAW 264.7 cells (mouse macrophage-like cells) treated with SC9 or P110 produced less mitochondrial ROS in the presence of LPS (Extended Data Fig. 11a,b). This indicates that SC9 blocks mitochondrial dysfunction in multiple cell types. Future studies will determine the mechanisms of disease pathophysiology and its inhibition *in vivo*.'

Comment 33: LPS is not a model of sepsis, as the preclinical studies of anti-TNF α compounds revealed to us; such agents were beneficial in many models using LPS, but notably disappointing in the more biologically poly- and mono-microbial preclinical models of sepsis and in human participants of clinical trials. I would encourage the removal of such terminology. Endotoxemia, rather is a model more appropriately described as TLR4 pathways.

RESPONSE 33: We agree and have now modified the text throughout the manuscript, accordingly. We thank the reviewer for this clarification. In addition, to improve the translational relevance of the LPS-induced endotoxemia study, we have also conducted an additional experiment with SC9 administered after symptom onset. Our data show that SC9 protection can still be noted even after inflammatory response has been mounted (see new Fig. 6j, k, l).

Modified text on p. 7: 'To assess whether SC9 can rescue mice from endotoxemia after symptom onset, we conducted an additional LPS-induced endotoxemia mouse experiment with SC9 delivered 4-hours after LPS administration (Fig. 6k protocol scheme). We found that at 4-hours post LPS treatment, inflammatory markers were universally elevated with 6/48 inflammatory markers up over 100x, 20/48 markers up over 10x, 7/48 markers up over 5x, and 14/48 markers up over 2x (see also Extended Data Fig. 11c). MIP-2, IL-6, CCL2, CCL4, CXCL1, and IL-22 were up over 200x from baseline indicating a strong induction of inflammation before treatment. IL-2 and IL-7 were not significantly elevated. Although rescue treatment 4-hours post-LPS treatment (0.3mg) caused only a smaller reduction in survival and body weight (Extended Data Fig. 11e, f), the mice exhibited significant reduction in disease symptoms as determined by the murine sepsis score and comparable to the benefit of SC9 pre-treatment (MSS; Fig. 6l vs. 6h); symptom severity plateaued right after SC9 administration (dashed line and blue arrow in Fig. 6l), resulting in a 45% decrease in the peak sepsis score (from 12.2 to 6.7; $p = 0.0029$; Fig. 6l). These results indicate that SC9 can still be efficacious when administered after the induction of inflammatory response.'

Comment 34: What was the rationale for restricting the studies to female Balb/c mice. In general, there are notable differences between male and female mice, with female mice, overall, exhibiting a greater 'resistance' to endotoxemia. In addition, the mice are particularly young; 7 weeks equates approximately to a teenage human. Confirmatory studies in male mice and the use of mice with comorbidities (e.g., obese, aged (>40 weeks)) are needed to support the conclusions made.

RESPONSE 34: This study is focused on the identification of the novel allosteric site in Drp1 and on a small molecule that interacts with it. The key observation and the goal of our study was to demonstrate that SC9 mimics P110. Indeed, we used only females to reduce variability; during

baseline observation, we noted significant fighting and wounds in the male cages so the study with the male mice was terminated. We now added this information in the limitations section. The sex-specific LPS-induced pathologies and the contribution of co-morbidity factors including age will be explored in future work.

Modified text on p. 10: 'Whereas the SC9 safety study was conducted in both male and female mice, the mouse endotoxemia studies were conducted in female mice only. Future studies will determine the sex-specific LPS-induced pathologies, the contribution of co-morbidity factors and their contribution to the efficacy of SC9 in endotoxemia models.'

Minor

Comment 35: Figure 4c: What was the duration of observation?

RESPONSE 35: Cells were imaged for 20 minutes. This information is now added to Fig.4 legend.

Comment 36: Figure 4d: How was the competing risk of death managed for the parameters monitored over time? I perceive that the mass data of Extended Figure 7 highlight the potential issue; at 40 hours, nearly all the mice receiving 0.325 mg have died; what does that mass value really represent?

RESPONSE 36: The reviewer probably refers to Fig. 6d. Body mass values are taken from the surviving vehicle- or SC9-treated mice; in other words, these data could have underrepresented the extent of mass loss due to survivor bias in the vehicle treated group, which experienced greater death than the SC9 treated group. This is now added to the text.

Modified text on p. 6: 'The effect of SC9 treatment vs. vehicle treatment on body mass of the mice could not be determined at time points later than 20 hours, because the most severely affected mice die and fall out of the analysis, resulting in survivor bias at LPS dose of 0.3mg and higher (Extended Data Fig. 10g,j).'

Comment 37: Toxicity of SC9: I do not consider body mass over 72 hours as a sufficient 'test' of toxicity, and though specific, is unlikely to be sensitive. There are several scales and/or assays specific to each organ system: Bone marrow (hematological analysis); Nervous system (cognitive, behavioral), Kidney (cystatin c, GFR), hepatic (biosynthesis). I am unable to identify the data of SC9 and MSS (Extended Data Figure 6) and draw note, like body mass, that the MSS was not developed nor validated to monitor and qualify toxicity.

RESPONSE 37: The reviewer is correct. Our original report on SC9's effect in naive mice only checked for overt toxicity and to rule out a confounding effect in the measures taken during the LPS challenge (mass and murine sepsis score). SC9 alone did not cause any clinical phenotype in these sepsis measures, an important control experiment. We have now added new data on SC9 effect in naive mice (Extended Data Fig. 9h-r; shown below). We used 6 mice per group with 3 male and 3 female mice. The length of the treatment (72hrs) matches that for the endotoxemia study and the dose of SC9 used was 50mg/Kg (2.5x greater than dose used in the LPS study). No effect was found on liver, kidney, or hematological functions.

Modified text on p. 6: 'Next, to determine if SC9 resulted in on-target or off-target toxicity in vivo, we conducted standard safety toxicology screening. Female and male 7-weeks old Balb/c mice were injected with 50mg/kg SC9 and observed for 3 days. Mice were then sacrificed and serum samples were screened for 11 biomarkers for hematologic, hepatic, renal, and musculoskeletal damage. No

significant difference in any of the clinical observations or serum measures were noted by an observer blinded to the experimental conditions (Extended Data Fig. 9h-r).'

REVIEWERS' COMMENTS

Reviewer #1 (Remarks to the Author):

The authors have attempted to address thus reviewers concerns over target site identification, by softening the language. While the modelling and mutagenesis is self-consistent, there is still a lingering doubt as to the exact identification of the target. In the revised text there is mention that MOE products the SWAG as the 3rd largest pocket. Did the authors look at mutant within the other pockets to determine effect?

Generally high Resolution MS / HPLC is not sufficient to confirm compound identity, even if purchased from a vendor. It is not unusual to find that the structure cited is not the structure supplied. Clean ¹H and ¹³C NMR combined with HPLC and MS, and ideally recapitulation of the study with newly synthesised (and characterised) material should be considered the gold standard with respect to compound identity and actual activity. The chemical synthesis of the SC compounds is relatively trivial and is recommended to validate in the absence of full spectroscopic characterisation.

The overall thrust of this work is worthy of publication, its impact would be better with 'solid' target identification. However, these 'probe' molecules require fully characterisation before entry into the public sphere.

Reviewer #2 (Remarks to the Author):

In the revised manuscript, Rios et al. satisfactorily address the concerns of the reviewers and have improved the manuscript significantly with several clarifications and experiments from Drp1^{-/-} cells showing that SC9 activity appears reliant on the presence of Drp1. They also provide Western blot analyses to show SC9 does not impact Drp1 expression.

Minor comments:

1. "and loss of function heterozygous variants can lead to severe neurological dysfunction in mice⁵⁶"

Not only in mice but also humans. e.g. PMID: 35914810, 36763487, 36795043

The author subsequently note that:

"In humans, de novo dominant-negative variants that inhibit function result in severe neuro-muscular diseases⁶²."

I note that some data support that human pathological variants might not all act in a dominant negative fashion (e.g. 35914810) and the authors might consider simply deleting "dominant-negative" from that sentence.

2. “BDY-P110 had an increased affinity for Drp1 in the presence of the non-hydrolysable GTP analog, GMPPCP ($K_d = 18\text{nM} \pm 23\text{nM}$) and GTP ($K_d = 58\text{nM} \pm 46\text{nM}$), as well as with GDP ($K_d = 219\text{nM} \pm 219$)”

Formally these affinity values should be reported as “apparent K_d ” values given the Drp1 assembly equilibria was presumably not accounted for in the fitting the data.

3. “, whereas Mdivi-150, another Drp1 inhibitor,”

I’m concerned that this statement will continue to fuel confusion in the field about whether Mdivi-1 inhibits Drp1 or not. The authors in their Response #19 state that:

“Unfortunately, we have not included Mdivi-1 in our GTPase assays, as its effect in vitro [reported by Nunnari (Dev Cell. 2008;14:193-204)] is not in dispute as far as we know; the debated effects of Mdivi-1 relate to the biological activity of the compound.”

The authors are correct that Mdivi-1 might be promiscuous and act in at least two manners that they correctly cite. However, they are incorrect about Nunnari and co-workers showing that Mdivi-1 inhibited Drp1 activity. In that work (Dev Cell. 2008;14:193-204), Mdivi-1 was identified using a phenotypic screen in yeast and, indeed, Mdivi-1 inhibited yeast Drp1 (Dnm1) GTP hydrolysis. In human cells, Nunnari and co-workers provide ample data that are consistent with inhibition of Drp1. However, at that time, recombinant Drp1 was not available and no enzyme assays with Drp1 were reported in that paper. Subsequently, Brian Polster and co-workers demonstrated Mdivi-1 activity against Complex I of the electron transport chain (as cited by Rios et al) . In enzyme assays reported in that work, the estimated $K_i > 1.2 \text{ mM}$ of Mdivi-1 against Drp1 isoform 1. It is possible that Mdivi-1 inhibits Drp1 on the mitochondria during the act of scission, which was not measured in the enzyme assays, however, several labs have tested Mdivi-1 in enzyme assays and in Drp1 knockout cells without the expected activity. My suggestion would be to alter the wording to reflect this uncertainty. Perhaps “a weak Drp1 inhibitor” would suffice.

4. “(The images for Fig. 5a were z-stacked, since the mitochondria are not all parallel to the bottom of the cell; a full cell z-stack captures polymerized Drp1 foci on all the mitochondria in the cell.)”

I appreciate this detail but could be moved to methods to not interrupt the flow of the manuscript.

Reviewer #3 (Remarks to the Author):

The authors have adequately addressed the identified concerns and sufficiently described the noted limitations.

REVIEWERS' COMMENTS

Reviewer #1 (Remarks to the Author):

The authors have attempted to address thus reviewers concerns over target site identification, by softening the language. While the modelling and mutagenesis is self-consistent, there is still a lingering doubt as to the exact identification of the target. In the revised text there is mention that MOE products the SWAG as the 3rd largest pocket. Did the authors look at mutant within the other pockets to determine effect?

Generally high Resolution MS / HPLC is not sufficient to confirm compound identity, even if purchased from a vendor. It is not unusual to find that the structure cited is not the structure supplied. Clean ¹H and ¹³C NMR combined with HPLC and MS, and ideally recapitulation of the study with newly synthesised (and characterised) material should be considered the gold standard with respect to compound identity and actual activity. The chemical synthesis of the SC compounds is relatively trivial and is recommended to validate in the absence of full spectroscopic characterisation.

The overall thrust of this work is worthy of publication, its impact would be better with 'solid' target identification. However, these 'probe' molecules require fully characterisation before entry into the public sphere.

Response: We thank the reviewer for the comments. We didn't look at mutants within the other pockets to determine the effect.

Reviewer is correct in pointing out that HRMS/HPLC is not sufficient to confirm compound identity. We have performed the full spectroscopic characterization of the SC compounds and are listed below. We will also add this to the supplementary material of the paper.

4,4'-((((4-methyl-1,3-phenylene)bis(azanediyl))bis(carbonyl))bis(azanediyl))bis(propane-3,1-diyl))bis(morpholin-4-ium) (SC1). ¹H NMR (600 MHz, dmsO) δ 8.31 (s, 1H), 7.68 (d, *J* = 2.3 Hz, 1H), 7.49 (s, 1H), 7.09 (dd, *J* = 8.2, 2.2 Hz, 1H), 6.90 (d, *J* = 8.3 Hz, 1H), 6.49 (t, *J* = 5.7, 5.7 Hz, 1H), 5.99 (t, *J* = 5.8, 5.8 Hz, 1H), 3.55 (t, *J* = 4.7, 4.7 Hz, 8H), 3.11 – 3.04 (m, 4H), 2.35 – 2.30 (m, 6H), 2.30 – 2.24 (m, 6H), 2.06 (s, 3H), 1.59 – 1.52 (m, 4H). ¹³C NMR (151 MHz, dmsO) δ 155.79, 155.70, 139.07, 138.63, 130.32, 119.73, 112.07, 110.72, 66.65, 56.33, 56.31, 53.83, 37.86, 37.81, 27.14, 17.65. HRMS (*m/z*): calc. for C₂₃ H₃₈ N₆ O₄ (M⁺) 462.2955, obs. 462.2951.

(4-amino-6-((4-chlorophenyl)amino)-1,3,5-triazin-2-yl)methyl 4-oxo-4-(3-phenyl-4,5-dihydro-1H-pyrazol-1-yl)butanoate (SC3). ¹H NMR (600 MHz, dmsO) δ 9.66 (s, 1H), 7.78 (d, *J* = 8.8 Hz, 2H), 7.75 – 7.70 (m, 2H), 7.46 – 7.42 (m, 3H), 7.29 (d, *J* = 8.9 Hz, 2H), 7.26 – 6.96 (m, 2H), 4.82 (s, 2H), 3.89 (t, *J* = 10.1, 10.1 Hz, 2H), 3.26 (t, *J* = 10.1, 10.1 Hz, 2H), 2.98 (t, *J* = 6.9, 6.9 Hz, 2H), 2.73 (t, *J* = 6.9, 6.9 Hz, 2H). ¹³C NMR (151 MHz, dmsO) δ 173.03, 172.67, 169.21, 166.99, 164.41, 156.84, 139.22, 131.73, 130.66, 129.19, 128.65, 126.96, 126.14, 121.73, 65.10, 44.49, 31.80, 28.80. HRMS (*m/z*): calc. for C₂₃ H₂₂ Cl N₇ O₃ (M⁺) 479.1473, obs. 479.1464.

***N*-(6-amino-1-benzyl-2,4-dioxo-1,2,3,4-tetrahydropyrimidin-5-yl)-2-((5,7-dimethyl-[1,2,4]triazolo[1,5-a]pyrimidin-2-yl)thio)-*N*-methylacetamide (SC9).** ¹H NMR (500 MHz, dmsO) δ 10.97 (s, 1H), 7.33 (td, *J* = 8.0, 7.8, 1.8 Hz, 2H), 7.24 – 7.18 (m, 3H), 7.16 (d, *J* = 7.8 Hz, 2H), 7.09 (s, 1H), 5.15 – 5.02 (m, 2H), 4.12 – 4.00 (m, 2H), 2.90 (s, 3H), 2.62 (s, 3H), 2.54 (s, 3H). ¹³C NMR (151 MHz, dmsO) δ 170.04, 165.89, 164.57, 159.86, 155.45, 153.39, 150.52, 146.68, 136.61, 128.98, 127.47, 126.33, 110.79, 94.37, 44.82, 35.37, 35.02, 24.85, 16.88. HRMS (*m/z*): calc. for C₂₁H₂₂N₈O₃S (M⁺) 466.1536, obs. 466.1528.

Reviewer #2 (Remarks to the Author):

In the revised manuscript, Rios et al. satisfactorily address the concerns of the reviewers and have improved the manuscript significantly with several clarifications and experiments from Drp1^{-/-} cells showing that SC9 activity appears reliant on the presence of Drp1. They also provide Western blot analyses to show SC9 does not impact Drp1 expression.

Minor comments:

1. “and loss of function heterozygous variants can lead to severe neurological dysfunction in mice⁵⁶”
Not only in mice but also humans. e.g. PMID: 35914810, 36763487, 36795043

The author subsequently note that:

“In humans, de novo dominant-negative variants that inhibit function result in severe neuro-muscular diseases⁶².”

I note that some data support that human pathological variants might not all act in a dominant negative fashion (e.g. 35914810) and the authors might consider simply deleting “dominant-negative” from that sentence.

Response: Done

2. “BDY-P110 had an increased affinity for Drp1 in the presence of the non-hydrolysable GTP analog, GMPPCP (K_d = 18nM ± 23nM) and GTP (K_d = 58nM ± 46nM), as well as with GDP (K_d = 219nM ± 219)”

Formally these affinity values should be reported as “apparent K_d” values given the Drp1 assembly equilibria was presumably not accounted for in the fitting the data.

Response: Done

3. “, whereas Mdivi-150, another Drp1 inhibitor,”

I’m concerned that this statement will continue to fuel confusion in the field about whether Mdivi-1 inhibits Drp1 or not. The authors in their Response #19 state that:

“Unfortunately, we have not included Mdivi-1 in our GTPase assays, as its effect in vitro [reported by Nunnari (Dev Cell. 2008;14:193-204)] is not in dispute as far as we know; the debated effects of Mdivi-1 relate to the biological activity of the compound.”

The authors are correct that Mdivi-1 might be promiscuous and act in at least two manners that they

correctly cite. However, they are incorrect about Nunnari and co-workers showing that Mdivi-1 inhibited Drp1 activity. In that work (Dev Cell. 2008;14:193-204), Mdivi-1 was identified using a phenotypic screen in yeast and, indeed, Mdivi-1 inhibited yeast Drp1 (Dnm1) GTP hydrolysis. In human cells, Nunnari and co-workers provide ample data that are consistent with inhibition of Drp1. However, at that time, recombinant Drp1 was not available and no enzyme assays with Drp1 were reported in that paper. Subsequently, Brian Polster and co-workers demonstrated Mdivi-1 activity against Complex I of the electron transport chain (as cited by Rios et al) . In enzyme assays reported in that work, the estimated $K_i > 1.2$ mM of Mdivi-1 against Drp1 isoform 1. It is possible that Mdivi-1 inhibits Drp1 on the mitochondria during the act of scission, which was not measured in the enzyme assays, however, several labs have tested Mdivi-1 in enzyme assays and in Drp1 knockout cells without the expected activity. My suggestion would be to alter the wording to reflect this uncertainty. Perhaps “a weak Drp1 inhibitor” would suffice.

Response: Done

4. “(The images for Fig. 5a were z-stacked, since the mitochondria are not all parallel to the bottom of the cell; a full cell z-stack captures polymerized Drp1 foci on all the mitochondria in the cell.)”

I appreciate this detail but could be moved to methods to not interrupt the flow of the manuscript.

Response: Done

Reviewer #3 (Remarks to the Author):

The authors have adequately addressed the identified concerns and sufficiently described the noted limitations.